# PANICL: MITIGATING OVER-RELIANCE ON SINGLE PROMPT IN VISUAL IN-CONTEXT LEARNING

## ABSTRACT

Visual In-Context Learning (VICL) uses input-output image pairs, referred to as in-context pairs (or examples), as prompts alongside query images to guide models in performing diverse vision tasks. However, VICL often suffers from over-reliance on a single in-context pair, which can lead to biased and unstable predictions. We introduce **PA**tch-based $k$-**N**earest neighbor visual **I**n-**C**ontext **L**earning (PANICL), a general training-free framework that mitigates this issue by leveraging multiple in-context pairs. PANICL smooths assignment scores across pairs, reducing bias without requiring additional training. Extensive experiments on a variety of tasks, including foreground segmentation, single object detection, colorization, multi-object segmentation, and keypoint detection, demonstrate consistent improvements over strong baselines. Moreover, PANICL exhibits strong robustness to domain shifts, including dataset-level shift (e.g., from COCO to Pascal) and label-space shift (e.g., FSS-1000), and generalizes well to other VICL models such as SegGPT, Painter, and LVM, highlighting its versatility and broad applicability.

## 1 INTRODUCTION

Facilitating large models' practical application of acquired knowledge through few-shot prompts is crucial for optimizing their performance. Large language models (LLMs), such as GPT (gpt; Achiam et al., 2023; Brown et al., 2020) and Gemini (Team et al., 2023), have achieved significant advancements across various challenging domains (Gonen et al., 2023; Wu et al., 2023; Wang et al., 2023d; Zhang et al., 2024b; Lee et al., 2025), largely due to their inherent capability for in-context learning (ICL). The ICL paradigm has also been extended to the field of computer vision (Bar et al., 2022; Wang et al., 2023b;c; Bai et al., 2024), referred to as visual in-context learning (VICL), enabling vision foundation models (VFMs) to tackle multiple tasks (e.g., segmentation, detection, and colorization) under the guidance of corresponding visual prompts without training.

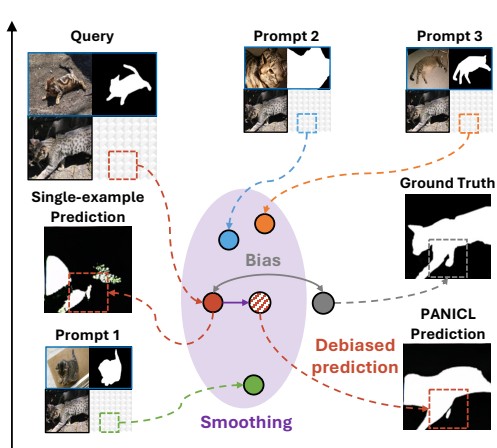

Figure 1: Each output token from MAE represents an assignment score over the VQGAN codebook. In single-example case, prediction may deviate from the ground truth. Averaging scores from multiple in-context pairs helps reduce this bias.

Visual prompting is one such approach. Pioneering work in this area includes MAE-VQGAN (Bar et al., 2022) and Painter (Wang et al., 2023b), which either combine MAE (He et al., 2022) with VQGAN (Esser et al., 2021) or are solely based on MAE. This VICL paradigm uses an input-output pair (termed an *in-context pair*) to demonstrate the desired output for the input image. The query image and an in-context pair are arranged in a four-cell grid canvas, called a *visual prompt*, enabling the model to generate the corresponding prediction.

Previous studies (Zhang et al., 2023; Sun et al., 2025) have explored methods for selecting effective in-context pairs given a query and have shown that in-context pairs similar to the query generally yield better performance. However, even the most similar in-context pair often results in inaccurate predictions (Xu et al., 2024; Suo et al., 2024), leading to the *training-needed* in-context pair selection (Zhang et al., 2023; Xu et al., 2024; Suo et al., 2024) to improve performance. These observations suggest that *VICL with a single in-context pair tends to over-rely on that pair, resulting in bias*. To the best of our knowledge, our work is the first to explicitly demonstrate that this single-example bias appears as deviations in the *assignment score space*. As illustrated in Figure 1, a prediction conditioned on a single in-context pair gives the detail of the output image of the pair, which are not necessarily close to the ground-truth.

In the NLP community, leveraging multiple in-context pairs has been shown to help LLMs better understand specific tasks and improve performance (Brown et al., 2020; Xu et al., 2023). Prior work (Zhou et al., 2024; Xu et al., 2023) suggests that aggregating information from multiple examples, rather than relying on a single one, can mitigate bias and improve the stability of predictions. In the context of VICL, naively importing this idea to visual prompting faces two key challenges: (1) Current VICL frameworks are often built on MAE variants and have strict input size limitations, making it difficult to include multiple in-context pairs. Some studies (Bar et al., 2022; Zhang et al., 2023), such as Large Canvas, have attempted to adopt a larger grid to place up to seven in-context pairs, resulting in a significant drop in performance. (2) Existing solutions, such as Feature Ensemble (Wang et al., 2023c) and Query Voting (Sun et al., 2025), are tailored to specific architectures or tasks and do not generalize well across different types of VICL models, including *discrete token-based models* (e.g., MAE-VQGAN (Bar et al., 2022)), *pixel-space models* (e.g., SegGPT (Wang et al., 2023c), Painter (Wang et al., 2023b)), and *autoregressive models* (e.g., LVM (Bai et al., 2024)).

To address these challenges, we propose **PA**tch-based $k$-**N**earest neighbor visual **I**n-**C**ontext **L**earning (**PANICL**). PANICL is built on MAE-VQGAN, where predictions are based on tokens (corresponding to image patches) in the VQGAN codebook, which are decoded into the output image by the pre-trained VQGAN decoder. We argue that each token's assignment scores over the codebook tend to over-rely on a single in-context pair. To mitigate this, PANICL constructs a *dynamic prompt pool* to store high-quality neighbors from different in-context pairs and performs smoothing on assignment scores identified as the $k$-nearest neighbors of the query. Conceptually, PANICL introduces a *patch-level, neighbor-aware assignment score smoothing* mechanism across multiple prompts, rather than operating only at the unconditioned feature or output pixel level as in existing ensemble or multi-example strategies. Although PANICL is described in the context of MAE-VQGAN, its design is *general* and can be adapted to other VICL models (e.g., SegGPT, Painter, and LVM) with minimal modifications.

**Contributions.** PANICL's design is motivated by the insight that biased predictions from single-prompt examples appear as deviations in the assignment score space. We propose a general framework to mitigate this bias by first introducing a *dynamic prompt pool* for constructing high-quality neighbors. Operating explicitly in the *assignment-score space*, which we identify as a general aggregation level for diverse VICL models, we then perform *patch-level, neighbor-aware smoothing* via similarity-based weighting. We conduct extensive experiments across diverse downstream tasks, including foreground segmentation, single object detection, colorization, multi-object segmentation, and key-point detection, demonstrating consistent improvements over strong baselines. Moreover, we verify PANICL's robustness to domain shifts and show its strong transferability across datasets (e.g., from COCO to Pascal) and generalizability to other VICL models such as SegGPT, Painter, and LVM, underscoring its versatility and broad applicability.

## 2 RELATED WORK

**In-Context Learning.** ICL is a groundbreaking approach within NLP, significantly enhancing the capabilities of LLMs like GPT-3 (Brown et al., 2020). This paradigm enables an autoregressive model to refine its performance without altering model parameters by using multiple predefined input-output pairs as prompts for specific tasks during inference. It provides a straightforward method for interacting with LLMs (Brown et al., 2020; Liu et al., 2022; Lu et al., 2022), aligning with cognitive processes in human decision-making (Winston, 1980), and facilitates the deployment of LLMs as readily available services (Dong et al., 2022). ICL has inspired innovative applications

across various domains (Chowdhery et al., 2023; Min et al., 2022; Kim et al., 2022), including mathematical reasoning (Wei et al., 2022), question-answering (Min et al., 2022; Press et al., 2023), and addressing compositional generalization challenges (An et al., 2023; Hosseini et al., 2022). ICL has also been explored beyond text, such as VICL (Bar et al., 2022; Wang et al., 2023c), multi-modal ICL (Alayrac et al., 2022; Huang et al., 2024; Li et al., 2025), and Speech ICL (Wang et al., 2023a).

**ICL in Computer Vision.** In computer vision, ICL extends its reach beyond textual data to non-textual data (Bar et al., 2022; Wang et al., 2023b;c; Bai et al., 2024). Unlike NLP's text-centric ICL, VICL involves defining and interpreting tasks through visual examples. Bar et al. (2022) pioneered the use of in-context pairs, along with a query image, to guide the model toward the desired inpainting result. This approach, which concatenates these elements into a single image, casts different tasks into a unified inpainting task. Painter (Wang et al., 2023b) proposed using the MAE architecture without VQGAN. SegGPT (Wang et al., 2023c) extended Painter to perform arbitrary image and video segmentation tasks using multiple visual examples. LVM (Bai et al., 2024) unlocked visual sentences by tokenizing images and using an autoregressive paradigm similar to LLMs.

The quality of the in-context images greatly affects performance (Zhang et al., 2023). Noting that similar input images in the prompt yield better outcomes, Zhang et al. (2023) proposed using CLIP (Radford et al., 2021) to find similar in-context pair. They also introduced contrastive learning to develop a similarity metric. Prompt-SelF (Sun et al., 2025) proposed a voting strategy to improve the VICL performance. InMeMo (Zhang et al., 2024a) introduced trainable perturbations to the in-context pair, which alter the features and potentially reduce reliance on the in-context pair. SegGPT (Wang et al., 2023c) employed spatial ensemble as a multi-example VICL technique similar to (Bar et al., 2022; Zhang et al., 2023), along with feature ensemble to average the transformer's intermediate features. LVM (Bai et al., 2024) proposed sequential modeling, allowing the model to accept up to seven in-context pairs simultaneously, greatly enhancing its capability.

Despite these advancements, over-reliance on a single example and misalignment with visual similarity (Suo et al., 2024; Xu et al., 2024) warrant further investigation, as they can introduce bias. Our study investigates how incorporating multiple examples can effectively mitigate this bias and be broadly applicable to VICL models. Moreover, a general method for different types of VICL models that leverages multiple examples also needs to be explored. In contrast to prior multi-example VICL methods such as Large Canvas (Bar et al., 2022), Query Voting (Sun et al., 2025), and Feature Ensemble (Wang et al., 2023c), which either arrange multiple examples onto a single canvas, ensemble only at the output pixel level, or simply average intermediate features, PANICL's patch-level, neighbor-aware assignment score smoothing explicitly targets over-reliance on a single prompt.

## 3 METHOD

### 3.1 PRELIMINARY: MAE-VQGAN

MAE-VQGAN is a pioneering work that first proposes VICL via inpainting (Bar et al., 2022). Let $x_q \in \mathbb{R}^{C \times H \times W}$ denote the query image that we wish to obtain the task output. An in-context pair for $x_q$ consists of an example input $x \in \mathbb{R}^{C \times H \times W}$ together with its ground-truth output $y \in \mathbb{R}^{C \times H \times W}$ (e.g., a segmentation result). The prompt is formed by concatenating $x_q$, $x$, and $y$ into a single image $c_q = [x, y, x_q, r] \in \mathbb{R}^{C \times 2(H+1) \times 2(W+1)}$,[1] where $[\cdot, \cdot, \cdot, \cdot]$ denote image concatenation to form a four-cell grid *canvas (a.k.a. prompt)*, the region $r \in \mathbb{R}^{C \times H \times W}$ is kept blank for the output $\hat{y}_q$.[2] The region $r$ is divided into $L$ image patches, forming a set $\mathcal{L} = \{r_l\}_{l=1}^{L}$ of patches $r_l$. MAE-VQGAN computes the *assignment score* vector of the patch region $r_l$ as:

$$\boldsymbol{s}_l = g_l(c_q) \quad \in \mathbb{R}^{|\mathcal{V}|}, \tag{1}$$

where $\mathcal{V}$ is the learned VQGAN codebook and $g_l$ is the MAE that computes the assignment score $\boldsymbol{s}_l$ for $r_l$. Element $s_{lv} \in \boldsymbol{s}_l$ is the score for the visual token $v \in \mathcal{V}$, which can also be interpreted as the probability $p(w_l = v | c_q)$ that the visual token assigned to $r_l$, denoted by $w_l$, is $v$. After finding the

---

[1] Following (Bar et al., 2022), we add a two-pixel gap between images.

[2] In (Bar et al., 2022), the arrangement of $x_q$, $x$, and $y$ is flexible; MAE-VQGAN can even accept more than one in-context pair. A mask is used to specify the region $r$ to be filled in. These details are omitted here for simplicity.

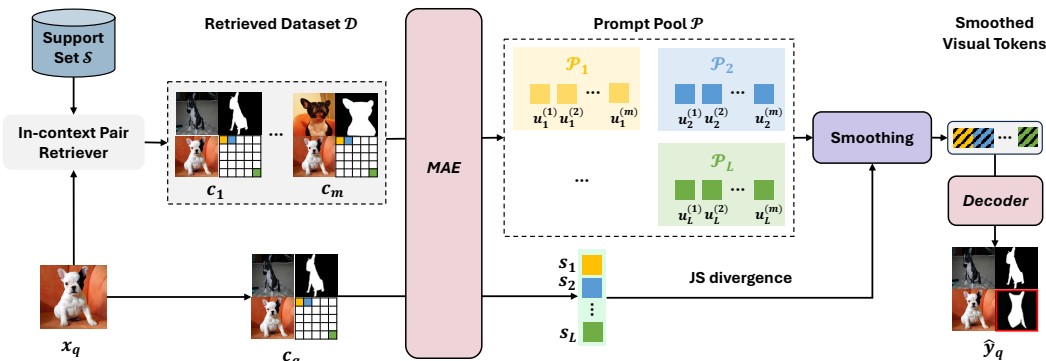

Figure 2: The overall pipeline of the proposed PANICL consists of two branches: *prompt pooling* (upper) and *query* (bottom). First, the *in-context pair retriever* is employed to select multiple in-context pairs, constructing the dataset $\mathcal{D}$. Subsequently, the patch-level assignment scores obtained by MAE are stored in the *prompt pool* $\mathcal{P}$. In the query branch, MAE is utilized to derive assignment scores for a given $x_q$. To mitigate over-reliance on a single prompt, *smoothing* based on the Jensen-Shannon (JS) divergence is applied. Finally, the smoothed visual tokens are input into the VQGAN decoder to generate the prediction $\hat{y}_q$.

visual token for $r_l$ by $w_l^\star = \arg\max_v s_{lv}$, VQGAN decoder $f$ generates the prediction $\hat{y}_q$, as:

$$\hat{y}_q = f(\{w_l^\star\}_{l=1}^L) \quad \in \mathbb{R}^{C \times H \times W}. \tag{2}$$

## 3.2 OVERVIEW OF PANICL

Let $\mathcal{S} = \{(x, y)\}$ denote a dataset of in-context pairs. Given a query image $x_q$, existing methods like (Zhang et al., 2023) typically identify the most similar image in $\mathcal{S}$ to $x_q$ and its ground-truth image as the in-context pair. Otherwise, multiple in-context pairs are formed using top-$m$ similar images and are packed into a single query image (Bar et al., 2022; Zhang et al., 2023), which results in sub-optimal performance. PANICL also uses multiple in-context pairs but forms a prompt for each pair with an *anchor* image in place of the query. The prompts are fed into MAE individually to obtain assignment scores for the anchors. These assignment scores are stored for smoothing.

Figure 2 shows the overall pipeline of PANICL, consisting of *prompt pooling* and *query* branches. These branches share the *in-context pair retriever* to identify the top-$m$ similar images from $\mathcal{S}$, resulting in the retrieved dataset $\mathcal{D} = \{(x_i, y_i) | i = 1, \ldots, m\}$. The query branch uses the pre-trained MAE-VQGAN encoder, $g_l$, to compute $s_l = g_l(c_q)$. In the prompt pooling branch, scores $u_l^{(i)} = g_l(c_i)$ are computed for pairs in $\mathcal{D}$ and an anchor $x_a$, i.e., $c_i = [x_i, y_i, x_a, r]$ for $i = 1, \ldots, m$, and stored in the *prompt pool* $\mathcal{P}$. PANICL then applies *assignment score smoothing* to $s_l$ to smooth out the details in $s_l$ using assignment scores in $\mathcal{P}$. By doing so, we expect to remedy over-reliance on the single most similar in-context pair $(x_1, y_1)$. Finally, the pre-trained VQGAN decoder $f$ generates a prediction $\hat{y}_q$ from the smoothed assignment scores.

Assignment score smoothing over $\mathcal{P}$ is the key component of PANICL. We argue that a prediction's over-reliance on the single input-output example $(x_1, y_1)$ as in the original MAE-VQGAN, introduces bias in $s_l$, potentially skewing the prediction $\hat{y}_q$ towards being similar to the corresponding patch in $y_1$ (see Figure 1). PANICL mitigates such bias by averaging out the individual examples with $\mathcal{P}$. The following sections detail our in-context pair retriever, prompt pool, and assignment score smoothing.

## 3.3 IN-CONTEXT PAIR RETRIEVER

Identifying a high-quality in-context pair for a specific query image is challenging yet crucial for better performance, as highlighted in (Zhang et al., 2023). To select the appropriate example, we initially employ an off-the-shelf feature extractor (e.g., the CLIP visual encoder (Radford et al., 2021)) to obtain visual feature map for both the query image $x_q$ and each in-context image $x \in \mathcal{S}$. These feature maps retain the spatial dimensions and the channel dimension. Following Sun et al. (2025), we flatten them into respective (1-dimensional) vectors for calculating similarity so that the similarity

takes the spatial information into account. We call it *pixel-level retrieval (PLR)*. We use $x$'s (and the corresponding $y$'s) that give the highest similarities to $x_q$. More specifically, the $i$-th in-context pair $(x_i, y_i)$ in $\mathcal{D}$ is $(x, y) \in \mathcal{S}$ whose $x$ gives the $i$-th largest value of dot similarity $e(x_q)^\top e(x)$, where $e(\cdot)$ denotes the feature extractor with flattening and $\ell_2$-normalization.

## 3.4 PROMPT POOL

For NLP tasks (Xu et al., 2023), prompts consist of *demonstrating pairs*, which define the downstream task by providing multiple input-output pairs, and an *anchor*, which serves as a query for the prompt and computes the classification score. Both the demonstrating pairs and anchor are drawn from a dataset of input-output pairs.

PANICL borrows this idea for handling multiple in-context pairs. Due to the fixed input image size of MAE-VQGAN, PANICL uses a single in-context pair (corresponding to a demonstrating pair) to form a prompt.[3] Prompt pooling is thus pivotal, as it aggregates assignment scores from multiple in-context pairs in $\mathcal{D}$ and anchors. These scores are stored for each patch, allowing the patch's position to be used in subsequent processes. There are multiple possible ways to form a prompt (e.g., selecting an in-context pair and an anchor from $\mathcal{D}$).

Let $(x_i, y_i)$ and $x_a$ denote the $i$-th in-context pair and an anchor, respectively. The prompt pool for patch $l$, denoted by $\mathcal{P}_l$, stores the corresponding assignment scores from multiple prompts as:

$$\mathcal{P}_l = \{\boldsymbol{u}_l^{(i)} = g_l([x_i, y_i, x_a, r]) | i = 1, \ldots, m\}. \tag{3}$$

By default, PANICL uses $x_q$ as anchor (i.e., $x_a = x_q$). We evaluate different combinations in our ablation study.

## 3.5 ASSIGNMENT SCORE SMOOTHING

Figure 1 shows that assignment scores $\boldsymbol{s}_l$ can be biased depending on the in-context pairs. Assignment score smoothing mitigates this bias by retrieving similar patches from $\mathcal{P}$ and averaging them.

To find the $k$-nearest neighbors for the assignment scores $\boldsymbol{s}_l$ associated with the prompt $c_q = [x_1, y_1, x_q, r]$, for which we wish to obtain the output, we opt for the symmetric Jensen-Shannon (JS) divergence (Lin, 1991) due to its superior performance.[4] Letting $\boldsymbol{a} = [a_1, \ldots, a_N]^\top$ and $\boldsymbol{b} = [b_1, \ldots, b_N]^\top$ denote two distributions, the JS divergence is defined as:

$$D_{\text{JS}}(\boldsymbol{a} \| \boldsymbol{b}) = \frac{1}{2}(D_{\text{KL}}(\boldsymbol{a} \| \boldsymbol{z}) + D_{\text{KL}}(\boldsymbol{b} \| \boldsymbol{z})), \tag{4}$$

where $\boldsymbol{z} = (\boldsymbol{a} + \boldsymbol{b})/2$ and $D_{\text{KL}}$ is the Kullback-Leibler (KL) divergence, given by:

$$D_{\text{KL}}(\boldsymbol{a} \| \boldsymbol{b}) = \sum_{i=1}^{N} a_i \log \frac{a_i}{b_i}. \tag{5}$$

We collect a set $\mathcal{A}_l = \{\hat{\boldsymbol{u}}_l^{(1)}, \ldots, \hat{\boldsymbol{u}}_l^{(k)}\}$ of $k$ similar assignment scores, where $\hat{\boldsymbol{u}}_l^{(i)} \in \mathcal{P}_l$ is the $i$-th nearest neighbor of $\boldsymbol{s}_l$ (i.e., $D_{\text{JS}}(\boldsymbol{u}_l^{(i)} \| \boldsymbol{s}_l)$ gives the $i$-th lowest distance in $\mathcal{P}_l$). We use the weighted sum of $u \in \mathcal{A}_l$ for average score smoothing, which is given by:

$$\hat{\boldsymbol{s}}_l = (1 - \alpha)\boldsymbol{s}_l + \alpha \sum_{u \in \mathcal{A}_l} u \cdot \gamma_l(\boldsymbol{s}_l, u), \tag{6}$$

where $\alpha$ is a constant that determines the contribution of smoothing. The weight $\gamma$ is determined based on the JS divergence as:

$$\gamma_l(\boldsymbol{s}_l, u) = \frac{\exp(-D_{\text{JS}}(\boldsymbol{s}_l, u)/\tau)}{\sum_{u'} \exp(-D_{\text{JS}}(\boldsymbol{s}_l, u')/\tau)}, \tag{7}$$

where the sum in the denominator is computed over all $u' \in \mathcal{A}_l$, and $\tau$ is the temperature scaling factor. After this smoothing, we use $\hat{\boldsymbol{s}}_l$ to predict $\hat{y}_q$.

---

[3]Again, MAE-VQGAN allows for multiple in-context pairs at the cost of image resolution. We chose to use only one in-context pair to retain details.

[4]To find $k$-nearest neighbors, we can also use patch similarity in the image domain or feature similarity, instead of assignment score similarity. Comparisons are provided in Section A.

## 4 EXPERIMENTS

### 4.1 EXPERIMENTAL SETUP

**Dataset and downstream tasks.** Following the MAE-VQGAN protocol (Bar et al., 2022), we evaluate PANICL on three downstream tasks. (1) *Foreground segmentation* (**FgSeg.**) aims to isolate prominent objects within the given query image using an in-context pair as an example. For this task, we utilize the Pascal-$5^i$ dataset (Shaban et al., 2017), which is divided into four distinct subsets, each comprising five classes. We report the mean Intersection over Union (mIoU) for all splits, along with the average mIoU across these splits. (2) *Single object detection* (**Det.**) seeks to evaluate a model's capacity to identify the bounding box for a prominent object in an image. We conduct our experiments using images and bounding boxes from the PASCAL VOC 2012 dataset (Everingham et al., 2010). As in (Bar et al., 2022), we use a subset of the dataset containing only a single object per image. For evaluation, we also use mIoU as the metric. (3) *Colorization* (**Color.**) evaluates the model's capability to colorize grayscale images using an in-context pair. In line with (Zhang et al., 2023; Xu et al., 2024), we randomly sample 50,000 images from the ILSVRC2012 training set (Russakovsky et al., 2015) as the support set and randomly select test images from the validation set. We use mean squared error (MSE) as the evaluation metric.

Beyond these tasks, we further consider more complex settings. For *multi-object segmentation* (**MOSeg.**) on ADE20K (Zhou et al., 2017), we evaluate PANICL with SegGPT (Wang et al., 2023c) and LVM (Bai et al., 2024). We use mIoU and mean pixel accuracy (mACC) as metrics, consistent with SegGPT. For LVM, in line with CoF (Zheng et al., 2025), we convert the predicted results into binary pixel masks and compare them with binary ground truth masks, measured by IoU and pixel accuracy (P-ACC). We also evaluate PANICL with Painter (Wang et al., 2023b) on the *multi-class keypoint detection* (**KpDet.**) task using the COCO dataset (Lin et al., 2014), reporting mean average precision (AP), since SegGPT often struggles with this task.

**Baseline methods.** For MAE-VQGAN, we follow the experimental settings in (Xu et al., 2024). The baselines include:

- **Multi-example VICL**: (1) *Large Canvas*: Following previous works (Bar et al., 2022; Zhang et al., 2023), we create a grid large enough to accommodate up to seven in-context pairs. (2) *Query Voting* (Sun et al., 2025): Predictions from different examples with the same query are ensembled by voting.

- **Single-example VICL**: We compare PANICL with various *training-free* methods that use only a single in-context pair, including MAE-VQGAN (Bar et al., 2022), UnsupPR (Zhang et al., 2023), VTV (Hojel et al., 2024), and prompt-SelF (Sun et al., 2025). The Pixel-level Retrieval (PLR) (Sun et al., 2025) is our baseline for comparison.

For SegGPT, we compare PANICL against Random, and Feature Ensemble (FE) (Wang et al., 2023c) combined with PLR (PLR + FE) as a stronger baseline. For LVM and Painter, we compare PANICL against Random, and PLR.

**Implementation details.** On MAE-VQGAN, for FgSeg., we treat each test sample as the query image and vary the number of in-context pairs $m$ from 2 to 7, retrieving examples from the training set $\mathcal{S}$ to form the prompt pool (i.e., $m = 2, \ldots, 7$). For Det., experiments are conducted in a single fold, and we similarly vary the number of in-context pairs from 2 to 7. The $\tau$ in Equation 7 is set to 1.0 for all tasks. We set $\alpha = 1.0$ for FgSeg. and Color., and $\alpha = 0.7$ for Det. task. The parameter $k$ is set to 5 (or $k = m$ if $m < 5$) for all tasks. We choose $x_q$ as the anchor (i.e., $x_a = x_q$ in Equation 3). For PANICL with voting strategy (PANICL$_{\text{w/ voting}}$), we follow the default settings of prompt-SelF (Sun et al., 2025). For SegGPT and Painter, we set $m = 2$, $\tau = 25$, and $\alpha = 0.5$ for all

Table 1: Performance on FgSeg., Det., and Color. for multi-example VICL and PANICL. The best scores are highlighted in **bold**.

| $m$ | Method | FgSeg. (mIoU ↑) | | | | | Det. | Color. |
|---|---|---|---|---|---|---|---|---|
| | | Fold-0 | Fold-1 | Fold-2 | Fold-3 | Mean | (mIoU ↑) | (MSE ↓) |
| $m = 1$ | Large Canvas | 22.79 | 27.91 | 24.20 | 21.84 | 24.18 | 18.25 | 0.97 |
| | PANICL | **36.42** | **38.47** | **34.56** | **34.12** | **35.89** | **28.08** | **0.63** |
| $m = 2$ | Large Canvas | 23.31 | 29.05 | 24.64 | 20.65 | 24.41 | 18.25 | 0.85 |
| | Query Voting | 35.68 | 39.12 | 35.92 | 33.25 | 36.01 | 25.15 | - |
| | PANICL | **37.37** | **40.11** | **37.68** | **34.49** | **37.41** | **29.37** | **0.61** |
| $m = 3$ | Large Canvas | 25.29 | 31.96 | 28.00 | 24.17 | 27.35 | 21.71 | 0.81 |
| | Query Voting | 36.63 | 38.99 | 36.17 | 32.68 | 36.12 | 27.93 | - |
| | PANICL | **37.43** | **40.48** | **37.91** | **35.42** | **37.43** | **29.31** | **0.60** |
| $m = 4$ | Large Canvas | 26.01 | 32.73 | 27.91 | 25.90 | 28.14 | 25.68 | 0.81 |
| | Query Voting | 37.45 | 39.84 | 37.06 | 33.35 | 36.93 | 26.73 | - |
| | PANICL | **38.18** | **40.63** | **37.82** | **35.02** | **37.91** | **29.20** | **0.60** |
| $m = 5$ | Large Canvas | 26.54 | 33.34 | 28.28 | 25.97 | 28.53 | 27.17 | 0.80 |
| | Query Voting | 37.39 | 39.65 | 36.71 | 32.46 | 36.55 | 28.19 | - |
| | PANICL | **38.00** | **40.42** | **38.02** | **34.70** | **37.79** | **29.27** | **0.60** |
| $m = 6$ | Large Canvas | 27.12 | 33.90 | 29.43 | 27.30 | 29.44 | 28.74 | 0.80 |
| | Query Voting | 37.90 | 39.88 | 37.22 | 33.01 | 37.00 | 27.50 | - |
| | PANICL | 37.78 | **40.53** | **38.15** | **34.63** | **37.77** | **29.75** | **0.60** |
| $m = 7$ | Large Canvas | 27.49 | 34.38 | 30.56 | 29.04 | 30.37 | **30.02** | 0.79 |
| | Query Voting | 37.68 | 39.70 | 36.83 | 32.48 | 36.67 | 28.16 | - |
| | PANICL | 37.60 | **40.20** | **37.90** | **34.53** | **37.56** | 29.17 | **0.60** |

Table 2: Performance on FgSeg., Det., and Color. tasks. The best scores in *training-free* for each fold are highlighted in **bold**. † means using the DINOv2 as the feature extractor aligned with Partial2Global. ★ indicates our reproduction results using the official code.

| | Venue | FgSeg. (mIoU ↑) | | | | | Det. (mIoU ↑) | Color. (MSE) ↓ |
| | | Fold-0 | Fold-1 | Fold-2 | Fold-3 | Mean | | |
|---|---|---|---|---|---|---|---|---|
| *Training-needed* | | | | | | | | |
| SupPR (Zhang et al., 2023) | NeurIPS'23 | 37.08 | 38.43 | 34.40 | 32.32 | 35.56 | 28.22 | 0.63 |
| SCS (Suo et al., 2024) | ECCV'24 | - | - | - | - | 35.00 | - | - |
| Partial2Global† (Xu et al., 2024) | NeurIPS'24 | 38.81 | 41.54 | 37.25 | 36.01 | 38.40 | 30.66 | 0.58 |
| *Training-free* | | | | | | | | |
| Random (Bar et al., 2022) | NeurIPS'22 | 28.66 | 30.21 | 27.81 | 23.55 | 27.56 | 25.45 | 0.67 |
| UnsupPR (Zhang et al., 2023) | NeurIPS'23 | 34.75 | 35.92 | 32.41 | 31.16 | 33.56 | 26.84 | 0.63 |
| VTV (Hojel et al., 2024) | ECCV'24 | 38.00 | 38.00 | 33.00 | 32.00 | 35.30 | - | - |
| PLR (Sun et al., 2025) | TIP'25 | 36.42 | 38.47 | 34.56 | 34.12 | 35.89 | 28.08 | 0.63 |
| PANICL ($m = 4$) | *Ours* | 38.18 | **40.63** | 37.82 | 35.02 | 37.91 | **29.20** | **0.60** |
| PANICL† ($m = 4$) | *Ours* | **38.63** | 40.44 | **39.50** | **35.89** | **38.62** | 28.85 | **0.60** |
| prompt-SelF★$_{\text{w/ voting}}$ (Sun et al., 2025) | TIP'25 | 41.54 | 44.45 | 39.85 | 35.92 | 40.44 | 29.83 | - |
| PANICL$_{\text{w/ voting}}$ ($m = 4$) | *Ours* | **43.85** | **45.29** | **42.09** | **36.19** | **41.86** | **31.05** | - |

FgSeg., MOSeg., and KpDet. tasks. For LVM on MOSeg., we set the maximum visual sentence length to 16, consisting of seven in-context pairs and one query as the prompt, leaving one slot for the output. We use two sequences for PANICL and set $\tau = 1.0$ and $\alpha = 0.8$.

## 4.2 COMPARISON WITH MULTI-EXAMPLE VICL BASELINES

Table 1 summarizes the scores. For FgSeg., the Large Canvas leads to significant performance degradation compared with PANICL. This aligns with results in (Zhang et al., 2023), indicating that increasing the number of in-context pairs in a single prompt is suboptimal for guiding the model. We attribute this degradation to the necessity of resizing the in-context pairs and a query, which results in a loss of image details. Query Voting achieves sub-optimal performance and is worse than PANICL. PANICL, on the other hand, demonstrates stable performance as the number of examples increases, and consistently outperforms the stronger baseline, Query Voting. We also observe that the model performance improves for the Large Canvas baseline as the number of examples increases. PANICL exhibits a similar trend when $m = 4$ but shows minor decreases as $m$ increases further. We hypothesize that, as $m$ grows, the off-the-shelf in-context pair retriever tends to introduce more globally similar but semantically misaligned examples (e.g., mismatched object position or scale) into the prompt pool. When such low-quality prompts are included, patch-level assignment score smoothing starts to mix useful and misleading signals, slightly degrading the final prediction. This observation is consistent with our failure case analysis in Section E.

In Det., the Large Canvas baseline performs progressively better as the number of in-context pairs increases, reaching its best result with 7 pairs (30.02%). In contrast, PANICL and Query Voting show inconsistent improvement. PANICL achieves the best result of 29.75% when $m = 6$. We consider that larger $m$ may over-smooth the prediction, reducing fidelity to the ground-truth image, which implies that choosing appropriate values for $m$ and $k$ is crucial for PANICL. Consistent with our observation in FgSeg., introducing too many prompts likely incorporates less relevant examples, resulting in a less-curated neighborhood that diminishes the smoothing benefits. It is also important to note that PANICL does not cause significant degradation in performance when dealing with multiple in-context pairs. We note that when $m = 2$, augmenting the simple prompt pool with just one additional sample yields a score improvement of 1.52 and 1.29 in both tasks compared to $m = 1$, confirming that simple assignment score smoothing at the patch level can effectively mitigate the bias caused by over-reliance on a single in-context pair.

For Color., Query Voting is not naturally suited for this task, demonstrating PANICL's versatility. The performance of Large Canvas increases continuously, achieving the best result at $m = 7$ with a score of 0.79. In contrast, PANICL remains stable as $m$ increases beyond two and consistently outperforms Large Canvas, achieving a score of 0.60.

## 4.3 COMPARISON WITH SINGLE-EXAMPLE VICL METHODS

Table 2 compares PANICL with *training-free* single-example VICL methods, including Random (Bar et al., 2022), UnsupPR (Zhang et al., 2023), PLR, and prompt-SelF (Sun et al., 2025). For reference,

we also include methods that require training (*training-needed*), including SupPR (Zhang et al., 2023), SCS (Suo et al., 2024), VTV (Hojel et al., 2024), and Partial2Global (Xu et al., 2024). PANICL outperforms all other *training-free* methods, with improvements of 2.02 and 1.12 improvements in the FgSeg. and Det., respectively, and 0.03 less in the Color. over the PLR. Compared with *training-needed* methods, PANICL also achieves the SOTA performance in the FgSeg. task. Additionally, with the same feature extractor (DINOv2 (Oquab et al., 2024)) in Partial2Global, PANICL even surpasses it by 0.22.

PANICL achieves the optimal result when $m = 4$. The superior performance compared to PLR across both downstream tasks suggests that PANICL's assignment score smoothing positively influences the predictions in the multi-example configuration. When using the voting strategy compared with prompt-SelF, PANICL achieves 1.42 and 1.22 gains for the FgSeg. and Det. tasks, respectively. These results demonstrate that simply smoothing the assignment score at the patch level can effectively mitigate the bias, leading to an effective and efficient method for VICL.

## 4.4 DOMAIN SHIFTS ANALYSIS

Domain shifts can occur in real-world applications, leading to variations in model performance compared to in-domain evaluations due to discrepancies in data distributions (Zhou et al., 2022). To examine how PANICL adapts to such domain shifts, we conduct experiments in two settings. For a dataset-level shift, we follow prior work (Sun et al., 2025; Zhang et al., 2024a) by deriving in-context pairs from COCO-$5^i$ and using query images from the Pascal-$5^i$ validation set (*COCO → Pascal*), *Pascal → Pascal* serves as the

Table 3: Results of dataset-level shift evaluation on PANICL and baselines.

| Setting | Method | Fold-0 | Fold-1 | Fold-2 | Fold-3 | Mean |
|---|---|---|---|---|---|---|
| *Pascal* | Large Canvas | 26.01 | 32.73 | 27.91 | 25.90 | 28.14 |
| → | PLR | 36.42 | 38.47 | 34.56 | 34.12 | 35.89 |
| *Pascal* | PANICL | 38.18 | 40.63 | 37.82 | 35.02 | 37.91 |
| *COCO* | Large Canvas | 23.68 | 29.64 | 24.15 | 24.08 | 25.39 |
| → | PLR | 33.94 | 37.45 | 32.45 | 33.21 | 34.26 |
| *Pascal* | PANICL | 35.14 | 39.22 | 37.32 | 33.48 | 36.29 |

in-domain reference. We employ $m = 4$ for Large Canvas and PANICL. The results are summarized in Table 3. For Large Canvas and PLR, mIoU decreases by 2.75 (-9.8%) and 1.63 (-4.5%), respectively, whereas PANICL decreases by 1.62 (-4.3%). For a label-space shift, we evaluate PANICL on FSS-1000 (Li et al., 2020) open-vocabulary segmentation task with 1k novel out-of-domain classes with $m = 4$, comparing to MAE-VQGAN and PLR in Table 4. PANICL demonstrates strong robustness across diverse domain shift scenarios.

## 4.5 PORTABILITY TO OTHER VICL MODELS AND TASKS

To evaluate the generalizability of PANICL to other VICL models and tasks, we assess it using SegGPT, LVM, and Painter in more complex scenarios, such as MOSeg. and KpDet., with pre-trained weights from the official repositories (see Tables 5 and 6). Note that for SegGPT, PLR is combined with Feature Ensemble (PLR + FE) as a stronger baseline. When applied to SegGPT, LVM, or Painter, PANICL achieves superior performance, demonstrating that it generalizes well to VICL models in pixel-space as well as to the autoregressive paradigm, with only minimal adjustments. While the numerical gains on the SegGPT backbone for the FgSeg. task are minor, they are consistent across different folds (see Table 17). This indicates that PANICL can effectively refine predictions even for advanced baselines in a training-free manner. Additional details regarding the transfer methods are provided in Section B.

## 4.6 VISUAL COMPARISON

We qualitatively compare PANICL with various baselines in Figure 3. The predictions indicate that PANICL maintains consistent performance across these tasks. Specifically, for FgSeg., Large Canvas and PLR often produce coarse or failed results, suggesting that the prompts do not effectively convey the task instructions. For Det., PANICL exhibits proficiency comparable to its performance on FgSeg., and notably shows more detail-oriented behavior when handling small objects. For Color., PANICL achieves the lowest MSE, indicating outputs closer to the GT. With SegGPT, PANICL yields improved results, producing complete and correct labels on both FgSeg. and MOSeg., demonstrating its versatility. With LVM, PANICL successfully identifies objects and produces detailed masks, whereas the baselines tend to leave more regions unidentified, which are visualized in black. With Painter, PANICL generates more accurate and detailed keypoints than the baselines. These visual examples suggest that using an appropriate number of in-context pairs can enhance performance by leveraging similar assignment-score distributions or intermediate features.

Table 4: FgSeg. results on FSS-1000.

| Method | FgSeg. mIoU ↑ |
|---|---|
| MAE-VQGAN | 58.30 |
| PLR | 58.67 |
| PANICL | **60.22** |

Table 5: Generalizability analysis of PANICL using SegGPT and LVM on FgSeg. and MOSeg. tasks.

| Method | FgSeg. (SegGPT) mIoU ↑ | MOSeg. (SegGPT) mIoU ↑ | mACC ↑ | MOSeg. (LVM) IoU ↑ | P-ACC ↑ |
|---|---|---|---|---|---|
| Random | 72.10 | 18.80 | 27.40 | 91.13 | 92.05 |
| PLR | 75.88 | 21.92 | 28.40 | 91.00 | 92.19 |
| PANICL | **76.13** | **21.97** | **28.43** | **91.78** | **92.73** |

Table 6: KpDet. results on COCO.

| Method | KpDet. AP ↑ |
|---|---|
| Painter | 71.8 |
| PLR | 72.1 |
| PANICL | **72.2** |

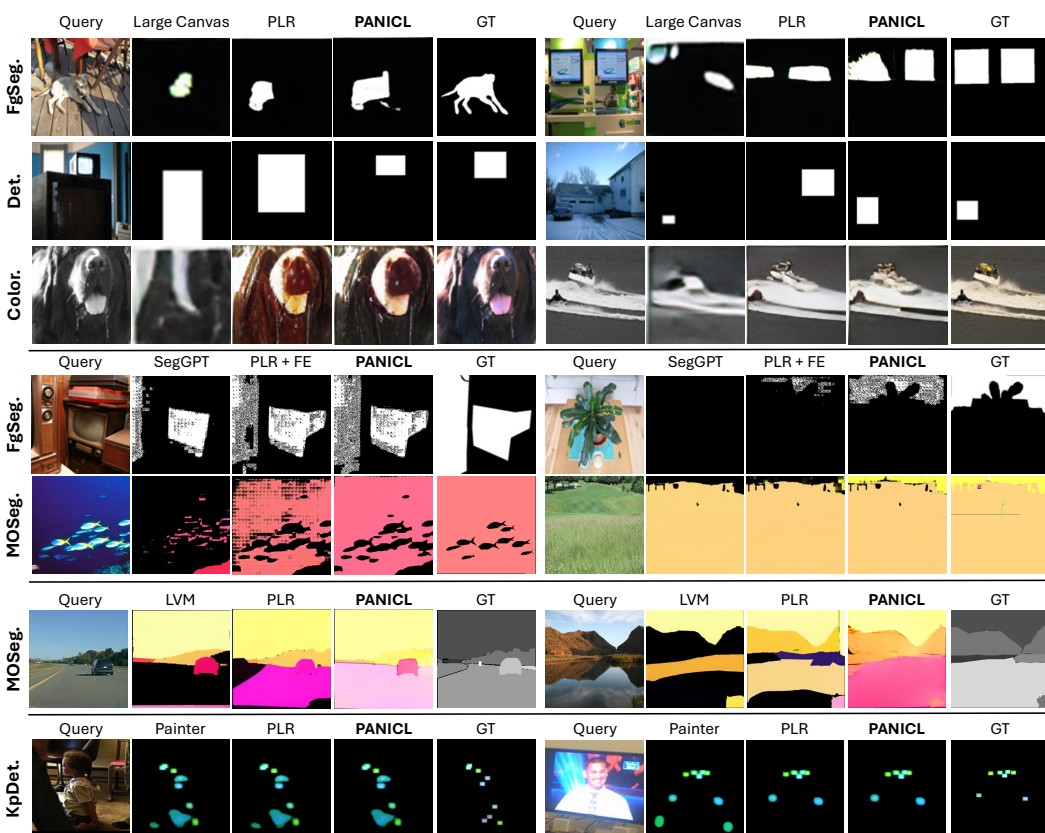

Figure 3: Visual examples comparing PANICL with baseline methods across FgSeg., Det., Color., MOSeg., and KpDet. tasks. We show two examples per row, including ground truth (GT) for reference. PANICL demonstrates enhanced VICL capability. Additional visual examples and failure cases are provided in Sections D and E.

## 4.7 FURTHER ANALYSIS OF PANICL

This section further explores PANICL using the optimal setting of $m = 4$ on MAE-VQGAN. Additional ablation studies and computational cost analyses are provided in Sections A and C.

**Different combinations of in-context pairs and anchors.** Being different from ICL for the NLP tasks (Xu et al., 2023; Zhou et al., 2024), PANICL uses arbitrary pairs in $\mathcal{D}$ as in-context pairs and also as anchors, while the query can also be an anchor. This arbitrariness makes it important to explore their choices, which alters prompt pool $\mathcal{P}_l$. We investigate different configurations of the in-context pair and anchor as variants of PANICL, namely $\mathcal{P}_l^{\text{rand}}$, $\mathcal{P}_l^{\text{seq}}$, and $\mathcal{P}_l^{\text{self}}$. The configuration adopted in PANICL (explained as the default setting in Section 3.4) is named $\mathcal{P}_l^{\text{q}}$.

Table 7: Results for different prompt pool configurations, with the best fold scores in **bold**.

| | Fold-0 | Fold-1 | Fold-2 | Fold-3 | Mean |
|---|---|---|---|---|---|
| $\mathcal{P}_l^{\text{rand}}$ | 21.19 | 28.08 | 24.37 | 22.76 | 24.10 |
| $\mathcal{P}_l^{\text{seq}}$ | 27.17 | 34.43 | 31.06 | 29.27 | 30.48 |
| $\mathcal{P}_l^{\text{self}}$ | 30.26 | 38.64 | 35.29 | 34.89 | 34.77 |
| $\mathcal{P}_l^{\text{q}}$ (Ours) | **38.18** | **40.63** | **37.82** | **35.02** | **37.91** |

Given a retrieved dataset $\mathcal{D}$ and query $x_{\text{q}}$, both $\mathcal{P}_l^{\text{rand}}$ and $\mathcal{P}_l^{\text{seq}}$ always uses $(x_1, y_1) \in \mathcal{D}$ as the in-context pair, where $x_1$ is the most similar to $x_{\text{q}}$. $\mathcal{P}_l^{\text{rand}}$ uses random input images in $\mathcal{D}$ as anchors,

while $\mathcal{P}_l^{\text{seq}}$ uses the input images $x_i \in \mathcal{D}$ for $i = 2, \ldots, m$ in a sequential order as anchors. $\mathcal{P}_l^{\text{itself}}$ uses the in-context pair's input image as anchor (i.e., $x_a = x_i$. PANICL's default setting, which uses each in-context pair in $\mathcal{D}$ with $x_q$ as the anchor, is denoted by $\mathcal{P}_l^q$.

Table 7 compares these variants. The $\mathcal{P}_l^{\text{rand}}$ variant yields the lowest performance (24.10%), indicating that a random anchor $x_a$ fails to produce useful association scores for debiasing. The $\mathcal{P}_l^{\text{seq}}$ variant also gets the lower performance (30.48%) compared to PANICL. These findings underscore the importance of smoothing scores with multiple different in-context pairs, and the anchors should be similar to the $x_q$. The $\mathcal{P}_l^{\text{itself}}$ variant outperforms $\mathcal{P}_l^{\text{rand}}$ and $\mathcal{P}_l^{\text{seq}}$, though $\mathcal{P}_l^q$ performs best overall. These results confirm that $\mathcal{P}_l^q$ in PANICL garners the most beneficial scores for debiasing.

**Retrieval from different spatial ranges of patches.** PANICL retrieves $k$-nearest neighbors at the patch level. We also explored the possibility of retrieving them from different spatial ranges of patches stored in the prompt pool $\mathcal{P} = \cup_l \mathcal{P}_l$, using the same setting as $\mathcal{P}_l^q$.

Specifically, for each query patch we define a local window of size (e.g., $2 \times 2$, $3 \times 3$, or $5 \times 5$) on the entire bottom-right $7 \times 7$ patch grid, anchored at the query patch. When this window would extend beyond the image boundary, we clamp its top-left corner to the last valid position so that the entire window remains inside the grid. Our PANICL and all-patch settings are special cases with window sizes of $1 \times 1$ (per-patch retrieval) and $7 \times 7$ (the maximum search window), respectively. The results are shown in Table 8.

Table 8: Comparison across spatial retrieval window sizes for PANICL.

|  | Fold-0 | Fold-1 | Fold-2 | Fold-3 | Mean |
|---|---|---|---|---|---|
| $2 \times 2$ | 37.34 | 39.98 | 37.37 | **35.02** | 37.40 |
| $3 \times 3$ | 37.01 | 39.76 | 37.08 | 34.70 | 37.14 |
| $5 \times 5$ | 36.73 | 39.66 | 36.95 | 34.56 | 36.98 |
| All-patch | 36.72 | 39.64 | 36.97 | 34.59 | 36.98 |
| PANICL | **38.18** | **40.63** | **37.82** | **35.02** | **37.91** |

PANICL achieves consistent improvements in mean mIoU, outperforming the all-patch and larger-window settings across the four folds. Additionally, as the window size increases, the performance degrades. These results demonstrate that smoothing at the patch level is highly effective in mitigating bias between the original score and the ground truth. While large-window and all-patch settings provide more options for $k$-nearest-neighbor retrieval, including spatially distant patches can introduce noise. Forcing the retrieved neighbors to lie at the same spatial locations yields better results.

**Static prompt pool.** We employ a dynamic prompt pool for each query, as its computational overhead is minor compared to the baselines (see Section C). To explore *cost-effective* alternatives, we constructed an offline, static, patch-level prompt pool using all samples in the support set $\mathcal{S}$. Specifically, since the query is unavailable during offline construction, each in-context example serves as its own anchor (i.e., $\mathcal{P}_l^{\text{self}}$), a strategy that yields the best performance in Table 7. The number of neighbors $k$ is set to 4. Table 9 reports the results.

Table 9: Comparison of static prompt pool.

|  | Fold-0 | Fold-1 | Fold-2 | Fold-3 | Mean |
|---|---|---|---|---|---|
| Static Pool | 35.71 | 39.05 | 36.62 | 33.42 | 36.20 |
| Dynamic Pool | **38.18** | **40.63** | **37.82** | **35.02** | **37.91** |

Our dynamic prompt pool consistently outperforms the static variant. We attribute this performance gap to the fact that the static pool cannot leverage query-anchored combinations; constructed without access to the query, it lacks the query-specific alignment that PANICL exploits. Nevertheless, the static prompt pool still benefits from our proposed assignment score smoothing, outperforming the single-prompt PLR baseline even without an in-context pair retriever.

## 5 CONCLUSION

In this study, we presented PANICL, a training-free framework that leverages multiple in-context pairs to mitigate the bias caused by relying on a single example in visual in-context learning. Through extensive experiments, we demonstrated that PANICL consistently improves performance across diverse downstream tasks, including foreground segmentation, single object detection, colorization, multi-object segmentation, and keypoint detection. These results validate our assumption that assignment scores from a single in-context pair are overly specialized, and that smoothing them with scores from neighboring pairs effectively reduces this bias. We further showed PANICL's robustness to domain shifts (e.g., COCO $\rightarrow$ Pascal, FSS-1000) and its adaptability to different VICL models such as SegGPT, Painter, and LVM, underscoring its versatility and broad applicability in VICL.

ETHICS STATEMENT

We adhere to the ICLR Code of Ethics. PANICL uses only publicly available datasets and official pre-trained weights under their respective licenses (e.g., Pascal-5$^i$, COCO, ADE20K, FSS-1000, ILSVRC2012; MAE-VQGAN, SegGPT, Painter, LVM). No human subjects or personally identifiable information were involved, and no re-identification was attempted. As a training-free method, we did not train any large models; all experiments relied solely on inference using publicly released weights. PANICL is intended for research and non-harmful applications, and we discourage any discriminatory or unsafe uses.

REPRODUCIBILITY STATEMENT

We have taken multiple steps to facilitate reproducibility. The paper specifies the datasets, splits, and evaluation protocols for all tasks (FgSeg., Det., Color., MOSeg., KpDet.) and provides implementation settings for PANICL in Section 4.1. Additional ablations and sensitivity analyses are provided in Section A, while transfer setups for other VICL models are described in Section B. Computational considerations are summarized in Section C. An anonymized code repository is included in the supplementary materials.

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

# A ADDITIONAL ABLATION STUDY AND ANALYSIS

This section gives the additional ablation study and analysis of PANICL on MAE-VQGAN.

**Temperature scaling factor $\tau$.** To explore PANICL's sensitivity to the temperature scaling factors $\tau$, we fixed the balance factor $\alpha$ at 1.0 and evaluated the performance across different values of $\tau$. Figure 4(a) summarizes the results. We observed a continue improvement on mean when $\tau$ increased. The performance plateaus when $\tau$ increases beyond 1.0, with the optimal performance (mean of 37.91%) achieved at $\tau = 1.0$. Beyond this point, increasing $\tau$ from 1 to 2 results in a slight decline in performance.

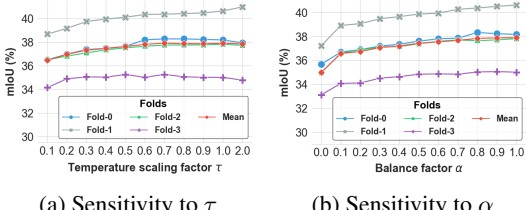

(a) Sensitivity to $\tau$      (b) Sensitivity to $\alpha$

Figure 4: Sensitivity analysis of PANICL.

**Balance factor $\alpha$.** We investigate the impact of varying $\alpha$. The results are shown in Figure 4(b). Incrementally raising $\alpha$ from 0 (original output) to 1.0 led to a steady improvement in the mean performance. The peak performance (Mean of 37.91%) was attained at $\alpha = 1.0$. There was a significant improvement when $\alpha$ increased from 0 to 0.1, indicating that even integrating surrounding samples with a minor balancing factor can lead to better performance.

Table 10: Ablation study of the $k$ ($m = 4$), with the best scores for each fold being highlighted in **bold**.

| $k$ | Fold-0 | Fold-1 | Fold-2 | Fold-3 | Mean |
|---|---|---|---|---|---|
| 1 | 36.32 | 38.57 | 36.37 | 33.95 | 36.30 |
| 2 | 37.11 | 39.52 | 37.38 | 34.66 | 37.17 |
| 3 | 37.93 | 40.11 | 37.51 | **35.22** | 37.69 |
| 4 | **38.18** | **40.63** | **37.82** | 35.02 | **37.91** |

**Number of $k$-nearest neighbors.** We investigate the ablations on $k$ (with $m = 4$), results are shown in Table 10. These results show that increasing $k$ improves performance consistently, supporting our claim of appropriately smoothing neighbors will mitigate the over-reliance on single prompt.

**Ablation study on weighted sum.** PANICL uses the softmax function for the weighted sum over different $k$-nearest neighbors in Equation 7. We conducted an ablation study on different settings for integrating neighbors. The results are also shown in the Table 11. "Nearest" refers to using the (single) nearest neighbor at the patch level, while "Average" refers to averaging the $k$-nearest neighbors without weighting. Apparently, using the nearest neighbor for smoothing leads

Table 11: Ablation study of the weighted sum, with the best scores for each fold being highlighted in **bold**.

| | Fold-0 | Fold-1 | Fold-2 | Fold-3 | Mean |
|---|---|---|---|---|---|
| Nearest | 36.32 | 38.57 | 36.37 | 33.95 | 36.30 |
| Average | 37.93 | **40.79** | 37.73 | 34.59 | 37.76 |
| PANICL | **38.18** | 40.63 | **37.82** | **35.02** | **37.91** |

to over-reliance on that specific patch, leading to same result with the PLR baseline. Meanwhile, simply averaging the $k$-nearest neighbors results in suboptimal outcomes, which can also demonstrate our suggestion that simply averaging the surrounding assignment scores can mitigate the bias. The superior performance of PANICL demonstrates that weighted smoothing of $k$-nearest neighbors based on JS divergence is the optimal choice.

**Comparison of different types of $Key$.** We proposed using the assignment score as the $key$ for finding $k$-nearest neighbors. We also conducted experiments with different settings for selecting $k$-nearest neighbors: $patch$ similarity and $feature$ similarity. $Patch$ similarity refers to the output image patch in pixel values produced by the VQGAN decoder, while $feature$ similarity refers to the intermediate features

Table 12: Comparison of different $key$ settings ($m = 4$).

| $Key$ | Fold-0 | Fold-1 | Fold-2 | Fold-3 | Mean |
|---|---|---|---|---|---|
| Patch | 34.30 | 38.47 | 34.84 | 31.18 | 34.70 |
| Feature | 36.61 | 40.58 | 36.81 | 33.58 | 36.90 |
| Score (PANICL) | **38.18** | **40.63** | **37.82** | **35.02** | **37.91** |

extracted before being passed into the final linear layer in MAE. Unlike the JS divergence used with assignment scores, we used $\ell_2$ distance to measure $patch$ and $feature$ similarities during the smoothing process. The results are shown in Table 12.

Regarding the different $key$ settings, we found that using the assignment score as the key yields the best performance for smoothing $k$-nearest neighbors. The patch-based key produced the worst results, likely due to the VQGAN decoder restoring the scores to pixel values, which causes minor differences to be mitigated by the VQGAN decoder. In the feature-based key, where intermediate

features are extracted before being transformed into codebook scores, the performance surpassed the patch-based key but did not match the score-based setting. Therefore, we demonstrate that building a prompt pool based on assignment scores provides PANICL with the best capability for debiasing.

**Comparison of KL and JS divergence.** We compared the performance of KL and JS divergence in selecting $k$-nearest neighbors ($m = 4$). The results are shown in Table 13.

Table 13: Comparison of KL and JS divergence on PANICL.

| Divergence | Fold-0 | Fold-1 | Fold-2 | Fold-3 | Mean |
|---|---|---|---|---|---|
| KL | 37.93 | **40.79** | 37.74 | 34.59 | 37.76 |
| JS (PANICL) | **38.18** | 40.63 | **37.82** | **35.02** | **37.91** |

The results demonstrate that JS divergence outperforms KL divergence across almost all folds. Specifically, KL can yield infinite values when the model's assignment scores of samples are excessively skewed, while JS exhibits greater stability and robustness, making it a more favorable choice for distance evaluation.

**Robustness to visually similar but semantically irrelevant in-context examples.** To evaluate this, we force the in-context pair retriever to return examples whose class labels differ from those of the query in each fold. We compare PANICL against the PLR baseline and report the mIoU on the FgSeg. task in Table 14. PANICL consistently outperforms the PLR baseline even under these challenging conditions, demonstrating that PANICL is robust to visually similar but semantically irrelevant examples.

Table 14: Comparison of PLR and PANICL on visually similar but semantically irrelevant examples.

| Method | Fold-0 | Fold-1 | Fold-2 | Fold-3 | Mean |
|---|---|---|---|---|---|
| PLR | 28.20 | 27.44 | 25.80 | 14.96 | 24.10 |
| PANICL | 30.62 | 30.40 | 28.83 | 18.42 | 27.07 |

**Analysis of retrieval quality when transferring across domains or object categories.** We investigate whether there is any degradation in retrieval quality when transferring across domains or object categories, and how this relates to VICL performance. We use Recall@5 to measure the fraction of queries for which at least one correct in-context example appears among the top-5 retrieved candidates. Using Recall@5, we compare within-domain retrieval (Pascal $\rightarrow$ Pascal) and cross-domain retrieval (COCO $\rightarrow$ Pascal), as shown in Table 15. In addition to the general domain shift, we observe a decline in Recall@5 on Fold-0 and Fold-1, whereas Fold-2 and Fold-3 remain stable. By aligning this with the corresponding per-fold performance, we find that folds with degraded Recall@5 correlate with larger drops in absolute mIoU across all methods (e.g., on Fold-0, PLR and PANICL decrease by 2.48 and 3.04, respectively). This confirms that retrieval quality influences VICL performance generally, rather than being a sensitivity unique to PANICL. In contrast, for folds without retrieval degradation, the performance drop is relatively small.

Table 15: Recall@5 (R@5) and mIoU for within-domain (Pascal$\rightarrow$ Pascal) and cross-domain (COCO$\rightarrow$ Pascal) retrieval. Red indicates decreased R@5, while blue indicates no decrease.

| | Pascal $\rightarrow$ Pascal | | | COCO $\rightarrow$ Pascal | | |
|---|---|---|---|---|---|---|
| | | mIoU $\uparrow$ | | | mIoU $\uparrow$ | |
| Fold | R@5 | PLR | PANICL | R@5 | PLR | PANICL |
| 0 | 0.90 | 36.42 | **38.18** | 0.87 | 33.94 | **35.14** |
| 1 | 0.90 | 38.47 | **40.63** | 0.89 | 37.45 | **39.22** |
| 2 | 0.86 | 34.56 | **37.82** | 0.87 | 32.45 | **37.32** |
| 3 | 0.84 | 34.12 | **35.02** | 0.84 | 33.21 | **33.48** |

Table 16: Per-class Recall@5 (R@5) and mIoU (for PLR and PANICL) on Pascal-5$^i$.

| | | | mIoU $\uparrow$ | | | | | mIoU $\uparrow$ | |
|---|---|---|---|---|---|---|---|---|---|
| Fold | Class (ID) | R@5 | PLR | PANICL | Fold | Class (ID) | R@5 | PLR | PANICL |
| 0 | aeroplane (01) | 1.00 | 49.82 | **51.21** | 2 | diningtable (11) | 0.58 | 21.17 | **22.94** |
| | bicycle (02) | 0.88 | 13.22 | **16.26** | | dog (12) | 0.95 | 46.99 | **49.13** |
| | bird (03) | 1.00 | 45.88 | **48.14** | | horse (13) | 0.94 | 46.60 | **47.04** |
| | boat (04) | 0.77 | 36.54 | **37.84** | | motorbike (14) | 0.89 | 47.42 | **47.97** |
| | bottle (05) | 0.78 | 26.51 | **28.03** | | person (15) | 0.88 | 32.37 | **33.76** |
| 1 | bus (06) | 0.92 | 57.60 | **61.56** | 3 | pottedplant (16) | 0.63 | 16.06 | **16.48** |
| | car (07) | 0.81 | 39.61 | **39.76** | | sheep (17) | 0.98 | 52.75 | **56.17** |
| | cat (08) | 0.98 | 48.45 | **52.53** | | sofa (18) | 0.76 | 23.01 | **23.49** |
| | chair (09) | 0.83 | 11.21 | **11.50** | | train (19) | 1.00 | 53.35 | **55.35** |
| | cow (10) | 0.98 | 46.52 | **50.34** | | tvmonitor (20) | 0.81 | 26.53 | **27.74** |

At the object category level, we report both Recall@5 and mIoU (for PLR and PANICL) for each class in Pascal-5$^i$. We observe that classes with low Recall@5 (e.g., *diningtable* with R@5 = 0.58,

*pottedplant* with R@5 = 0.63) exhibit lower absolute mIoU compared to classes with high Recall@5 (e.g., *aeroplane*, *train* with R@5 = 1.00) for both methods. This confirms a positive correlation between retrieval quality and mIoU across classes. Crucially, however, PANICL consistently improves over PLR across all classes, including those with poor retrieval quality. For instance, even on challenging classes such as *diningtable* and *pottedplant*, PANICL provides non-trivial gains over PLR ($21.17 \rightarrow 22.94$ and $16.06 \rightarrow 16.48$ mIoU, respectively). These observations suggest that while a stronger retriever could further boost absolute performance, our assignment score smoothing remains beneficial and robust across a wide range of retrieval qualities.

**Applicability of assignment score smoothing on pixel-space models.** We use $\ell_2$ distance to smooth neighbors at the patch level for pixel-space VICL models. To further demonstrate the effectiveness of our main claim on *assignment score smoothing*, we instead use JS divergence on SegGPT for the FgSeg. task. The results are shown in Table 17. Both distance measures (with $\ell_2$ and with JS) consistently improve over PLR + FE, and the differences between $\ell_2$ and JS are very small for each fold. This suggests that assignment score smoothing is still beneficial for pixel-space VICL models, even though the underlying representation is not a discrete codebook.

Table 17: Comparison of PLR + FE and PANICL with different distance measures ($\ell_2$ vs. JS).

| Method | Fold-0 | Fold-1 | Fold-2 | Fold-3 | Mean |
|---|---|---|---|---|---|
| PLR + FE | 69.44 | 77.26 | 79.06 | 77.77 | 75.88 |
| PANICL ($\ell_2$) | 69.68 | **77.50** | **79.29** | **78.05** | **76.13** |
| PANICL (JS) | **69.76** | 77.44 | 79.27 | 77.95 | 76.11 |

**Practical guidance.** While PANICL introduces several hyper-parameters, potentially raising concerns about the need for careful tuning, we find that extensive per-task tuning is unnecessary in practice. Using MAE-VQGAN as a representative backbone, simple defaults suffice. For fine-grained tasks (e.g., segmentation and colorization), using smaller $m$ and $k$ (e.g., 3 or 4) with default temperature $\tau$ and balance factor $\alpha$ (e.g., $\tau=1.0$, $\alpha=1.0$) effectively leverages multiple prompts. For coarse-grained tasks (e.g., detection), using slightly larger $m$ and $k$ (e.g., 5 or 6) and a reduced $\alpha$ (e.g., 0.7 or 0.8) yields strong performance. Crucially, we directly apply these settings to new datasets (e.g., COCO and FSS-1000) without modification, demonstrating PANICL's robustness to domain shifts and novel out-of-domain classes.

## B    DETAILS ON TRANSFERRING PANICL TO OTHER VICL MODELS

**Pixel-space VICL models (SegGPT and Painter).** Unlike MAE-VQGAN, the intermediate outputs of SegGPT and Painter are features rather than probability distributions. SegGPT employs Feature Ensemble (FE) (Wang et al., 2023c) at each attention layer by averaging multiple examples. Accordingly, we adopt a similar approach, using the $\ell_2$ distance to smooth neighbors at the patch level for each attention layer, instead of the JS divergence. We set $m = 2$, $\tau = 25$, and $\alpha = 0.5$.

**Discrete token-space autoregressive model (LVM).** For LVM, since it leverages a VQGAN encoder to convert images into discrete tokens, similar to the encoding method used in MAE-VQGAN, we extend PANICL to such an autoregressive LVM pipeline. Specifically, multiple in-context examples combined with the same query are treated as a visual sentence and fed into the LVM to generate output distributions over the pre-trained VQGAN codebook. In our implementation, we adopt a maximum input size of 16 images, including seven in-context pairs and one query as the prompt, leaving one slot for the output image per sentence. We set the number of visual sentences to two for PANICL, each constructed with different in-context pairs retrieved by the in-context pair retriever. The same weighted-sum smoothing strategy as in PANICL is applied to aggregate the two output distributions for a given query. The resulting smoothed distribution is finally decoded by the VQGAN decoder to produce the final output.

## C    COMPUTATIONAL COST

This section provides a detailed analysis of PANICL in terms of inference time and the computational cost of the retriever.

**Inference time and overhead.**    Table 18 compares inference speeds.  Compared with the Large Canvas and PLR baselines, PANICL spends more time on prompt pool construction (Pool). Although this introduces additional latency, PANICL enhances the quality of downstream tasks. We also quantify the overhead of Pool, and benchmark Pool per prompt in Table 19. These results demonstrate

Table 18: Inference time comparison on the Pascal-$5^i$ dataset.

| | Large Canvas | PLR | PANICL | | |
| --- | --- | --- | --- | --- | --- |
| | | | Pool | Infer | Total |
| Time [s] | 0.083 | 0.059 | 0.098 | 0.082 | 0.180 |

that the pool construction incurs only 0.025 s per prompt and modest memory overhead ($\approx$ 4 MB per prompt). Therefore, the additional latency and memory footprint of Pool are negligible in real-time applications, especially given the substantial performance gains.

**Computational cost of retriever.**    We follow SupPR (Zhang et al., 2023) in precomputing all images embeddings offline. To quantify the online overhead, we benchmark for support sets of size $\mathcal{S}$ in Table 20.  Even for $\mathcal{S} = 5,000$, the online retrieval cost is less than 12 µs per query, which is negligible.

Table 19:  Resource usage of Pool per prompt.

| | Infer (s/prompt) | GPU (MB/prompt) | Cache (MB/prompt) |
| --- | --- | --- | --- |
| Pool | 0.025 | 3.3 | 3.9 |

Table 20: Computational cost of retriever.

| $\mathcal{S}$ | Feature Extraction (ms / image) | Similarity Calculation (µs / query) | GPU (MB / query) |
| --- | --- | --- | --- |
| 1,000 | 10.1 | 4.0 | 34.8 |
| 5,000 | 9.9 | 11.3 | 34.8 |

## D    ADDITIONAL VISUAL EXAMPLES

**More visual examples across different downstream tasks.**    We provide additional visual examples for reference in FgSeg. (see Figure 5), Det. (see Figure 6), Color. (see Figure 7), MOSeg. (see Figures 8 and 9), and KpDet. (see Figure 10), respectively.

Across the three tasks on MAE-VQGAN, the Large Canvas baseline often produces predictions lacking fine-grained details due to the downsampling of in-context pairs, even though it reduces over-reliance on a single pair. In contrast, PLR generates more detailed outputs but remains highly sensitive to the chosen in-context pair. PANICL outperforms both baselines by producing more detailed and less biased results through assignment score smoothing. Specifically, in FgSeg., PANICL can generate a correct and detailed mask for the query image, whereas baseline methods often fail to predict or produce inaccurate masks. In Det., PANICL can consistently generate a correct and detailed bounding box, while baselines often produce boxes that are misplaced or incorrectly sized. In Color., PANICL outputs colors that closely match the ground truth, whereas baselines often generate distorted results that differ significantly.

For SegGPT on MOSeg., PANICL produces more accurate and detailed masks, while baseline methods tend to miss or misidentify objects. For LVM on MOSeg., PANICL shows its effectiveness by recognizing more objects, whereas baselines leave many areas unidentified, shown in black in the output. For KpDet. using Painter, PANICL generates more accurate keypoints by leveraging multiple examples to mitigate the effect of incorrect predictions, which baselines cannot achieve.

The visual examples demonstrate that leveraging prompt pooling to adjust the model's initial assignment scores effectively suppresses noise and enhances prediction details. Moreover, we observe that PANICL allows for precise control over the spatial positioning of outputs, enabling significant improvements by refining assignment scores. Compared to the baselines, PANICL produces predictions that align more closely with the ground truth. These results support the hypothesis that using an appropriate number of in-context pairs with similar assignment scores can lead to improved task performance. This qualitative evidence reinforces our core assumption: *assignment scores are biased*

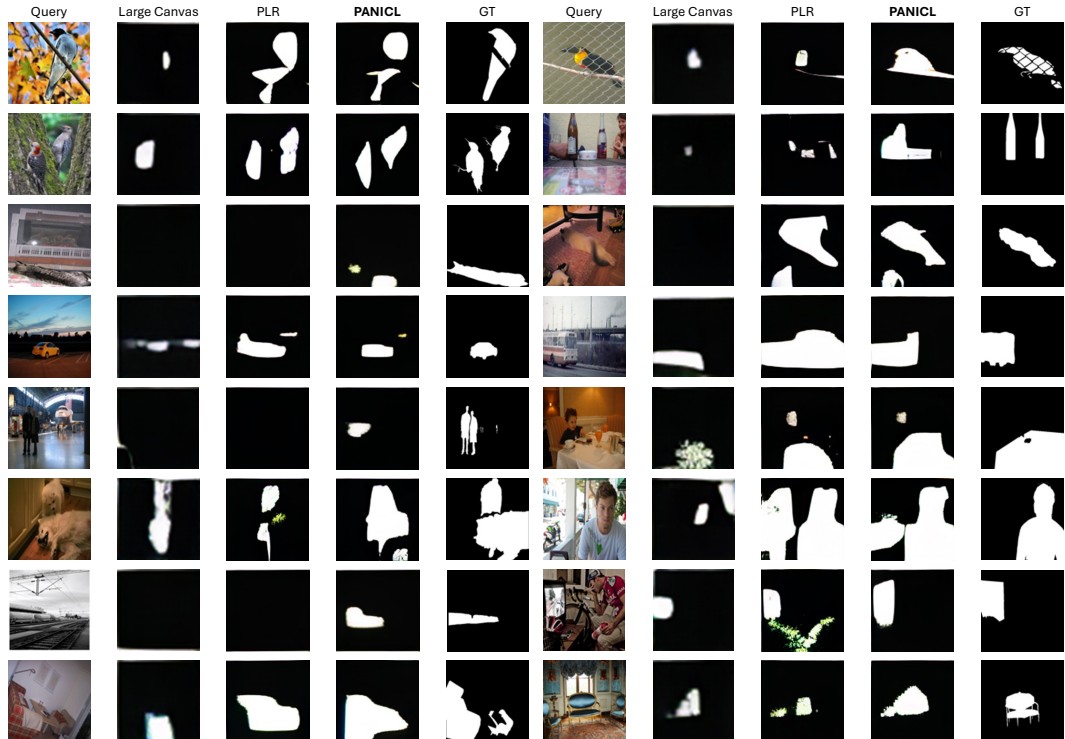

Figure 5: Additional visual examples from the foreground segmentation task comparing Large Canvas, PLR, and PANICL. Each row contains two examples. The results are arranged from left to right as follows: Large Canvas, PLR, **PANICL**, and ground truth (GT). PANICL demonstrates enhanced capability for VICL.

*due to over-reliance on a single in-context pair, and a simple averaging of assignment scores across multiple in-context pairs can effectively mitigate this bias.*

**Visual examples on more downstream tasks.** We further demonstrate PANICL's effectiveness on additional tasks using MAE-VQGAN, such as Edge Detection (**ED**) and Inpainting (**IP**), as shown in Figure 11. These visual examples highlight PANICL's robustness and versatility across diverse downstream tasks.

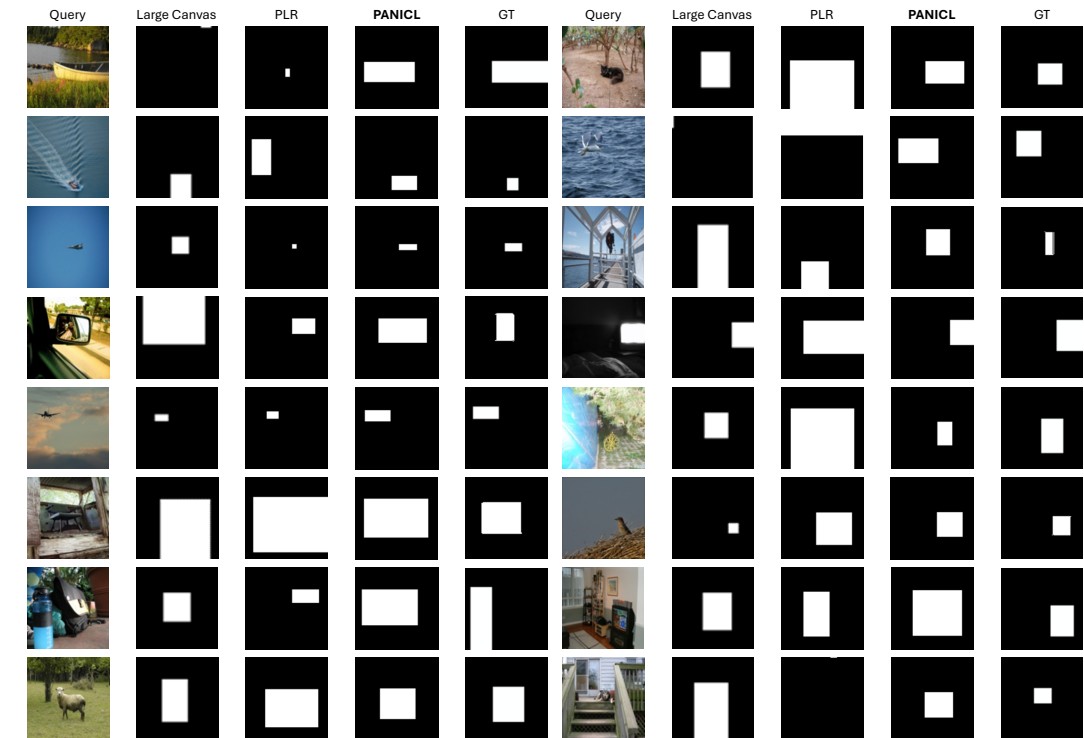

Figure 6: Additional visual examples from the single object detection task comparing Large Canvas, PLR, and PANICL. Each row contains two examples. The results are arranged from left to right as follows: Large Canvas, PLR, **PANICL**, and ground truth (GT). PANICL demonstrates enhanced capability for VICL.

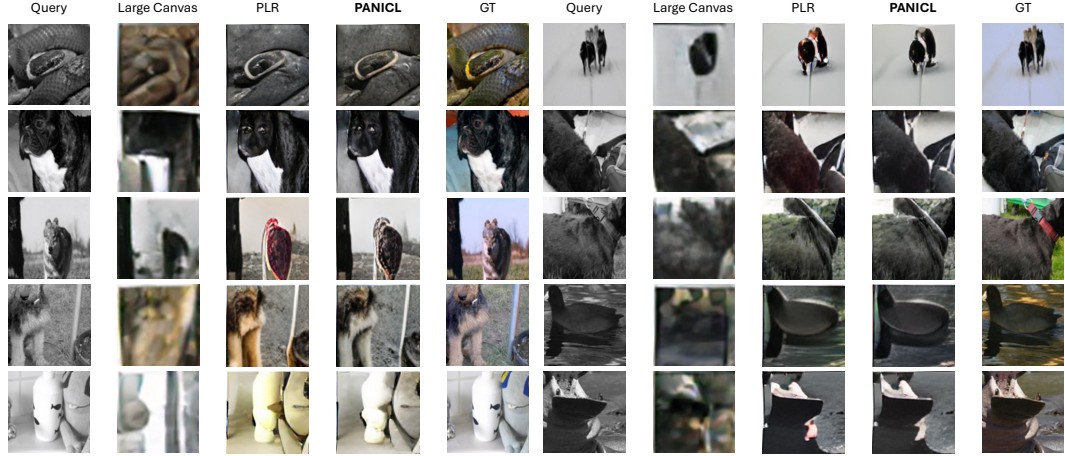

Figure 7: Additional visual examples from the colorization task comparing Large Canvas, PLR, and PANICL. Each row contains two examples. The results are arranged from left to right as follows: Large Canvas, PLR, **PANICL**, and ground truth (GT). PANICL demonstrates enhanced capability for VICL.

## E    FAILURE CASE ANALYSIS

We present several failure cases in Figure 12. In some instances, reducing reliance on a single in-context example can lead to worse results than those from single-example predictions. Additionally, using multiple retrieved examples based solely on off-the-shelf models may negatively impact VICL

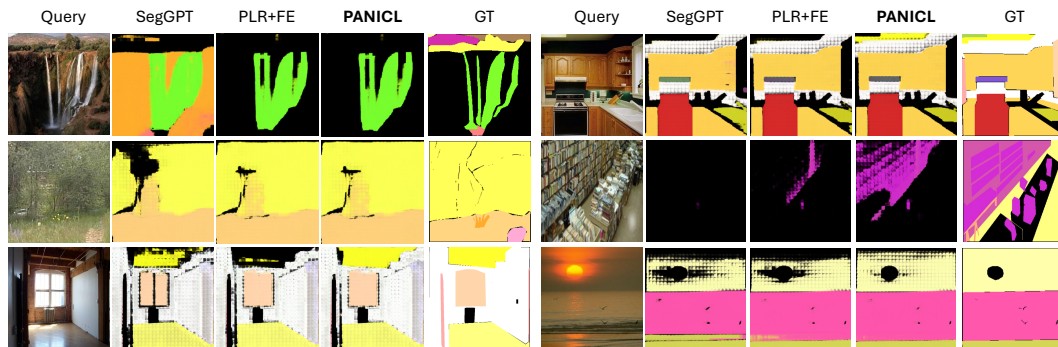

Figure 8: Additional visual examples from the MOSeg. comparing SegGPT, PLR combined with FE (PLR + FE), and **PANICL** on SegGPT. Each row contains two examples. PANICL demonstrates superior capability in VICL.

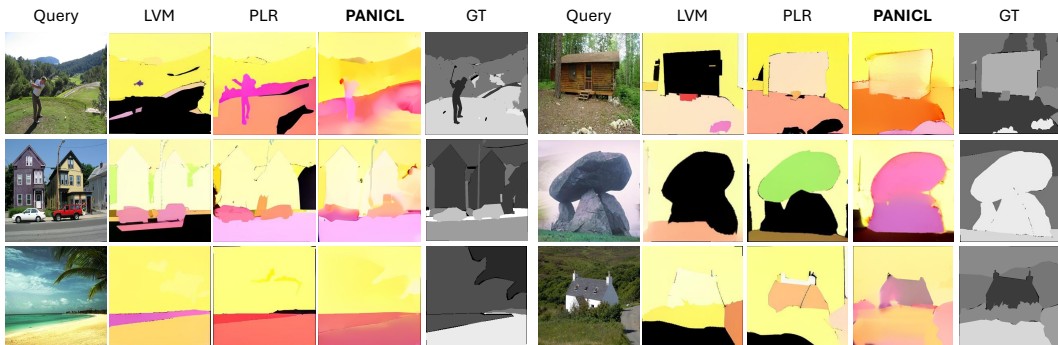

Figure 9: Additional visual examples from the MOSeg. task comparing LVM, PLR, and **PANICL** on LVM. Each row contains two examples. Black regions in the predicted results indicate unidentified objects. PANICL demonstrates superior capability in VICL by successfully identifying the correct objects.

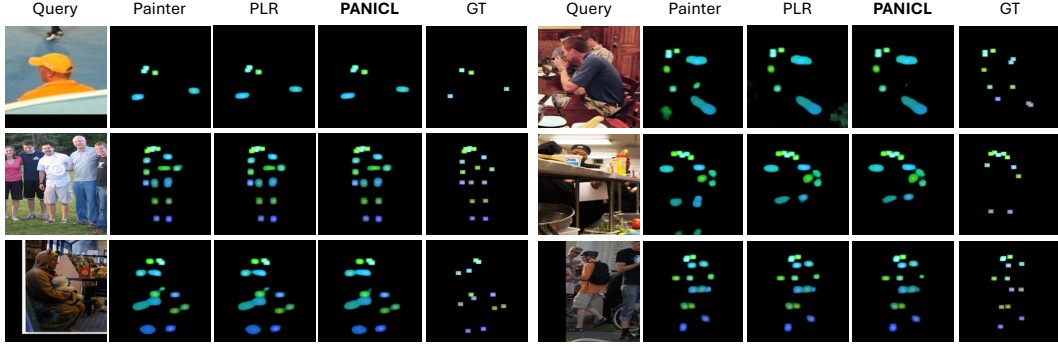

Figure 10: Additional visual examples from the KpDet. task comparing Painter, PLR, and **PANICL** on Painter. Each row contains two examples.

performance in some cases. We conduct an example-wise quantitative analysis of these failure cases, as summarized in Table 21. Specifically, for each failure case, we input the query with different in-context pairs (from top-1 to top-4 retrieved by the in-context pair retriever) into the model separately, reporting the results. In most cases, only the Top-1 example consistently provides a reliable match. Lower-ranked examples often degrade performance, as they are retrieved by off-the-shelf retriever based on global appearance, such as similar background texture or object categories, without guaranteeing object position, scale, or scene context alignment.

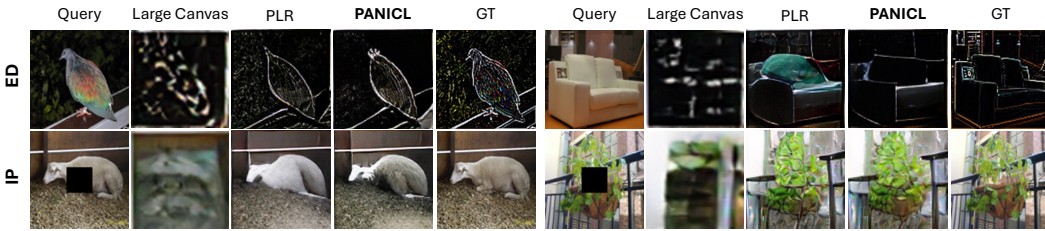

Figure 11: Visual examples on Edge Detection (**ED**) and Inpainting (**IP**) tasks.

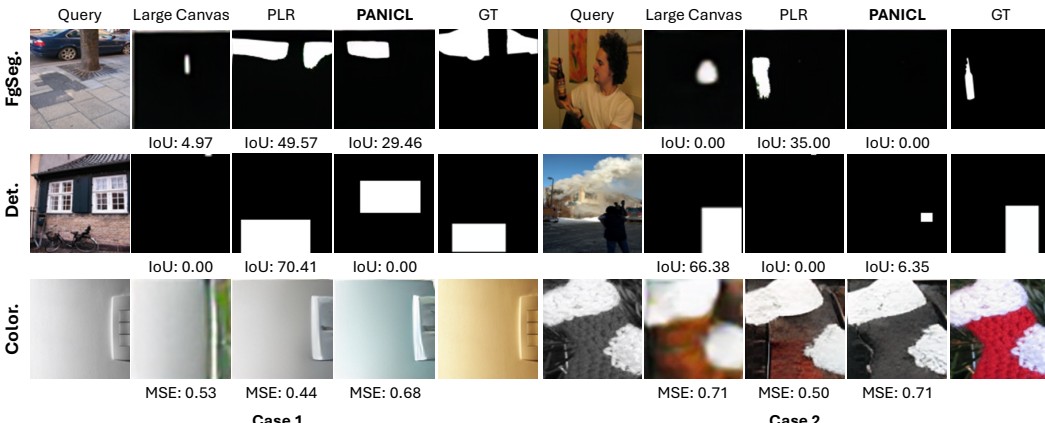

Figure 12: Failure cases across diverse downstream tasks, comparing Large Canvas, PLR, and **PANICL**. Each row shows two examples (Case 1 and Case 2) with their corresponding metrics below.

Qualitative Insight: We analyze the visualizations of each failure case at the example-wise. In FgSeg. Case 2, the Top-2 and Top-3 examples contain the correct object category, the objects are located differently or appear at different scales, confusing the VICL model. Similar misalignments are observed in Det. and Color., where example–query mismatches in layout or scene disrupt the output. Therefore, the main cause of failure cases lies in inaccurate example–query alignment, where retrieved examples often mislead the model due to mismatches in object position, scale, or scene context. These issues are primarily due to the limited discriminative power of the off-the-shelf feature extractor, which tends to prioritize global appearance over fine-grained semantics.

Table 21: Performance comparison across tasks for Top-1 to Top-4 cases retrieved by the in-context pair retriever.

| Task | Case | Top-1 | Top-2 | Top-3 | Top-4 |
|---|---|---|---|---|---|
| FgSeg. (mIoU) | 1 | 49.57 | 0.47 | 26.97 | 10.30 |
|  | 2 | 35.00 | 0.00 | 0.00 | 0.00 |
| Det. (mIoU) | 1 | 70.41 | 28.69 | 0.00 | 0.00 |
|  | 2 | 0.00 | 0.00 | 0.00 | 28.73 |
| Color. (MSE) | 1 | 0.44 | 0.79 | 0.68 | 0.65 |
|  | 2 | 0.50 | 0.71 | 0.69 | 0.63 |

## F   DISCUSSION AND FUTURE DIRECTIONS

As with other visual prompting methods, while PANICL can enhance the performance of VICL models, it remains constrained by the capacity of the underlying VICL backbone. Current VICL models also exhibit a sizable gap relative to SOTA task-specific models, indicating substantial room to develop more advanced, more general, and more robust VICL models. Although PANICL shows improvements with the multi-example configuration, its performance does not consistently increase as the number of prompts grows. Hyperparameters can be optimized for specific tasks or VICL models. However, they can also be chosen experimentally or based on prior experience. The results in Table 1 demonstrate that PANICL remains robust across a broad range of values ($m \in [2, 7]$), showing only a 0.5-point and 0.6-point variation in mIoU for FgSeg. and Det., respectively, and a 0.01-point variation in MSE for Color. Given these minor fluctuations, carefully tuning $m$ is generally

unnecessary. In practice, we find that $m = 3$ or $4$ works well for fine-grained tasks (e.g., FgSeg. and Color.), while a larger value is preferable for coarse-grained tasks (e.g., Det.). Importantly, PANICL effectively addresses over-reliance on a single prompt, which is a core contribution of this work. We believe PANICL offers an efficient and effective way to improve VICL performance.

From a conceptual perspective, PANICL can be interpreted as performing patch-level, neighbor-aware smoothing of assignment scores (or features) across prompts. For each patch, the assignment score conditioned on a single in-context pair can be viewed as a biased estimate of the ideal score aligned with the ground truth. Aggregating scores from $k$-nearest neighbors mitigates this single-prompt bias in the assignment score space. We consider a fully rigorous theoretical analysis to be an exciting direction for future research.

Regarding failure cases, PANICL currently relies on an off-the-shelf in-context pair retriever for retrieval and smoothing. Future research directions include: (1) designing or fine-tuning a better feature extractor for VICL to further improve retrieval quality, and (2) dynamically selecting the number of $m$ or $k$ based on the characteristics of each query or task to enhance adaptability and performance.

## G THE USE OF LARGE LANGUAGE MODELS (LLMS)

We used LLMs for grammar checking.

