# OpenReview forum: "PANICL: Mitigating Over-Reliance on Single Prompt in Visual In-Context Learning"
_ICLR.cc/2026/Conference — Submitted to ICLR 2026_

### Official Review · Reviewer_rqcE · 2025-10-29

**Soundness:** 3
**Presentation:** 3
**Contribution:** 2
**Rating:** 4
**Confidence:** 3

**Summary:**

The paper tackles the task of Visual In-Context Learning (VICL). VICL often suffers from over-reliance on a single in-context pair. To mitigate this issue, the paper introduces PANICL, a training-free method that smooths token-level assignment scores across multiple retrieved visual examples (prompts) using Jensen–Shannon divergence-based weighting. Finally, the authors test the approach on three downstream tasks and several benchmarks.

**Strengths:**

The paper tackles an important problem and is fairly well written. The idea is simple and based on the experimental results seems quite effective. It may have broad applications on multiple down-stream tasks.

**Weaknesses:**

My main concern is related to the novelty which seems a bit minimal. While it's an interesting idea, I feel like the proposed smoothing is a minor change on existing ensemble or multi-example strategies. I also feel that the “Model-agnostic” claim feels a bit exaggerated since the overall performance depends heavily on the architecture and retrieval quality even though technically can be applied to different models. However, as the experiments show, in some cases the performance improvements are small and often within noise margins.

Minor
* Tab 1 - wrong bolded value for m=6, 7 for fold-1

**Questions:**

Can you please share some more insights on why the performance degrades with increased m? Since the performance peaks with few samples, I believe that choosing those samples is very important so I think the performance might be very tightly coupled with the performance of the retrieval system. So, can you please elaborate on how different retrieval performance affects the proposed method?

---

> ### Author Response · Authors · 2025-11-21
> **Response to Reviewer rqcE (Part 1, Q1)**
>
> We thank Reviewer ``rqcE`` for the thoughtful comments and valuable suggestions, which have significantly contributed to enhancing the clarity and depth of our work. Below, we address the questions and comments in detail.
> ***
> >**Q1: Performance degradation when increasing the number of prompts $m$.**\
> *Can you please share some more insights on why the performance degrades with increased $m$?…*
>
> **A1:** Thank you for your insightful question. This behavior arises because as $m$ increases, the off-the-shelf in-context pair retriever tends to include more irrelevant examples in the prompt pool, which dilutes the effectiveness of smoothing. This phenomenon is evidenced in Table 8, where applying smoothing over larger windows or all patches leads to performance degradation. Furthermore, as noted in our failure case analysis in Appendix Section E, when the retriever introduces low-quality or misleading features, the added prompts can confuse the VICL model, ultimately harming performance.
>
> In the revised paper, we have added a detailed analysis in Section 4.2 explaining this performance degradation with larger $m$. Please kindly refer to it.

---

> ### Author Response · Authors · 2025-11-21
> **Response to Reviewer rqcE (Part 2, Q2)**
>
> >**Q2: Sensitivity to retrieval quality.**\
> *…performance might be tightly coupled with the performance of the retrieval system… how different retrieval performance affects the proposed method?*
>
> **A2:** We thank the reviewer for this important question. As a retrieval-based VICL method, the absolute performance of PANICL is naturally influenced by the quality of the in-context retriever. As suggested, we therefore investigate two main questions: (1) how retrieval quality varies across settings and affects performance, and (2) whether PANICL still provides consistent gains over single-prompt VICL under varying retrieval quality.
>
> First, we explicitly quantify retrieval quality using Recall@5 (R@5) and report the corresponding mIoU results as shown below. We compare within-domain retrieval (Pascal $\rightarrow$ Pascal) and cross-domain retrieval (COCO $\rightarrow$ Pascal). In addition to the general domain shift, we observe a decline in Recall@5 on Fold-0 and Fold-1, whereas Fold-2 and Fold-3 remain stable. When we compare the corresponding per-fold performance, we find that folds with degraded Recall@5 correlate with larger drops in absolute mIoU across all methods (e.g., on Fold-0, PLR and PANICL decrease by 2.48 and 3.04, respectively). This confirms that retrieval quality influences VICL performance generally, rather than being a sensitivity unique to PANICL. In contrast, for folds without retrieval degradation, the performance drop is relatively small.
>
> |Fold|Pascal $\rightarrow$ Pascal|||COCO $\rightarrow$ Pascal|||
> |:-:|:-:|:-:|:-:|:-:|:-:|:-:|
> |   |R@5|PLR (mIoU)|PANICL (mIoU)|R@5|PLR (mIoU)|PANICL (mIoU)|
> |0|0.90|36.42|**38.18**|0.87|33.94|**35.14**|
> |1|0.90|38.47|**40.63**|0.89|37.45|**39.22**|
> |2|0.86|34.56|**37.82**|0.87|32.45|**37.32**|
> |3|0.84|34.12|**35.02**|0.84|33.21|**33.48**|
>
> To analyze this more finely, we report per-class R@5 and mIoU (for PLR and PANICL) on Pascal-5$^i$. We see that classes with low R@5 (e.g., diningtable with R@5 = 0.58, pottedplant with R@5 = 0.63) indeed have lower absolute mIoU than classes with high Recall@5 (e.g., aeroplane, train with R@5 = 1.00), for both PLR and PANICL. This confirms that retrieval quality and mIoU are positively correlated across classes. Crucially, however, PANICL improves over PLR in every single class, including those with poor retrieval quality. For example, even on challenging classes such as diningtable and pottedplant, PANICL still provides a non-trivial gain over PLR (21.17 $\rightarrow$ 22.94 and 16.06 $\rightarrow$ 16.48 mIoU, respectively).
> |Fold|Class (ID)|R@5|PLR (mIoU)|PANICL (mIoU)|Fold|Class (ID)|R@5|PLR (mIoU)|PANICL (mIoU)|
> |:-:|:-|:-:|:-:|:-:|:-:|:-|:-:|:-:|:-:|
> |0|aeroplane (01)|1.00|49.82|**51.21**|2|diningtable (11)|0.58|21.17|**22.94**|
> | |bicycle (02)|0.88|13.22|**16.26**| |dog (12)|0.95|46.99|**49.13**|
> | |bird (03)|1.00|45.88|**48.14**| |horse (13)|0.94|46.60|**47.04**|
> | |boat (04)|0.77|36.54|**37.84**| |motorbike (14)|0.89|47.42|**47.97**|
> | |bottle (05)|0.78|26.51|**28.03**| |person (15)|0.88|32.37|**33.76**|
> |1|bus (06)|0.92|57.60|**61.56**|3|pottedplant (16)|0.63|16.06|**16.48**|
> | |car (07)|0.81|39.61|**39.76**| |sheep (17)|0.98|52.75|**56.17**|
> | |cat (08)|0.98|48.45|**52.53**| |sofa (18)|0.76|23.01|**23.49**|
> | |chair (09)|0.83|11.21|**11.50**| |train (19)|1.00|53.35|**55.35**|
> | |cow (10)|0.98|46.52|**50.34**| |tvmonitor (20)|0.81|26.53|**27.74**|
>
> Finally, in Table 14 in Appendix Section A, we further conduct an experiment where the retriever is forced to return semantically misaligned examples (i.e., retrieved in-context examples must have different class labels from the query). Even under this unfavorable setting, PANICL consistently outperforms the single-prompt PLR baseline across all folds. This shows that PANICL remains beneficial and relatively robust, rather than being overly brittle or tightly coupled to a particular retriever.
>
> In summary, our analyses show that retrieval quality indeed affects absolute performance (as expected for single-prompt VICL methods and PANICL), but across datasets, classes, and even severe retrieval settings, PANICL consistently provides improvements over single-prompt VICL.
>
> In the revised paper, we have added the mIoU results to Tables 15 and 16 in Appendix Section A and incorporated the above analysis. Please refer to them.

---

> ### Author Response · Authors · 2025-11-21
> **Response to Reviewer rqcE (Part 3, Q3, Q4, and Q5)**
>
> >**Q3: Novelty and distinction from existing strategies.**\
> *My main concern is related to the novelty… the proposed smoothing is a minor change on existing ensemble or multi-example strategies.*
>
> **A3:** We thank the reviewer for the constructive feedback regarding the novelty of our work. While PANICL is implementation-friendly, we respectfully disagree that it is merely a minor change to existing strategies. Instead, it is a general framework based on a novel perspective: **to the best of our knowledge, our work is the first to explicitly demonstrate that single-prompt VICL exhibits systematic bias in the assignment score space, and that smoothing in this specific space effectively mitigates such bias.**
>
> Our core methodological contribution goes beyond simple ensembling. We formulate a **patch-level, neighbor-aware assignment score smoothing** mechanism and a **dynamic prompt pool** strategy. Below, we clarify how these components distinguish PANICL from existing approaches:
>
> - **Distinction from multi-example VICL methods (Large Canvas, Query Voting).**\
> **Large Canvas** operates at the input level, concatenating examples into a single canvas. This necessitates resizing (loss of detail) and lacks intermediate smoothing. **Query Voting** operates at the output pixel level, which limits its applicability primarily to binary foreground segmentation. **In contrast, PANICL** identifies the **assignment score space** as a general and effective aggregation level. By keeping examples at original resolution and smoothing assignment scores, PANICL avoids resizing artifacts (unlike Large Canvas) and generalizes to diverse tasks like multi-object detection, colorization and keypoint detection (unlike Query Voting).
>
> - **Distinction from ensemble methods (Feature Ensemble).**\
> **Feature Ensemble** simply averages features across prompts, treating all examples equally, and is often tailored to specific architectures. **In contrast, PANICL** introduces a **neighbor-aware** mechanism. We explicitly enforce spatial locality and use similarity-based weighting (JS divergence or $\ell_2$) to let compatible neighbors contribute more. This is not a heuristic tweak but a principled method to reduce variance in the assignment-score space.
>
> **In summary,** PANICL is not just a combination of known techniques. It systematically addresses the "Why" (bias in assignment scores) and the "How" (dynamic pool + neighbor-aware smoothing). We therefore consider the systematic formulation and validation of this general framework to be a meaningful and non-trivial methodological contribution beyond a minor modification of existing heuristics.
>
> In the revised paper, we have further highlighted these distinctions in Sections 1 and 2.
> ***
> >**Q4: "Model-agnostic" claim and small, noisy improvements.**\
> *…the "Model-agnostic" claim feels a bit exaggerated… in some cases the performance improvements are small and often within noise margins.*
>
> **A4:** Thank you for raising this concern regarding the *"model-agnostic"* claim. As suggested, we have revised the description to *"general"* in Section 1 for better precision.
>
> Regarding the results, we acknowledge that the performance gain on foreground segmentation might appear modest in quantitative metrics (Table 5) compared to qualitative observations (Figure 3). However, we respectfully argue that these improvements are robust and **not attributable to noise:**
> - **Consistency across folds:** While the numerical gains on the SegGPT backbone are incremental, **they are consistent across all folds (Table 17 in Appendix Section A).** This consistency indicates that PANICL reliably refines predictions even for advanced baselines in a training-free manner.
> - **Bridging the gap to training-based methods:** For keypoint detection with Painter, our PANICL results are on par with the learnable prompting strategy reported in the Painter paper, while remaining completely training-free. This suggests that assignment score smoothing can close much of the gap to learned prompt optimization.
>
> In the revised paper, we have incorporated this analysis into Section 4.5.
> ***
> >**Q5: Incorrect bolded values in Table 1.**\
> *Tab 1 – wrong bolded value for m=6, 7 for fold-1.*
>
> We thank the reviewer for carefully checking Table 1 and pointing out this mistake. The boldface for the results was, indeed, incorrectly assigned. In the revised version, we have corrected the bolded entries to match the true best-performing values in Table 1.
>
> Last but not least, we sincerely thank Reviewer ``rqcE`` again for the valuable comments provided during this review.

---

### Official Review · Reviewer_guEZ · 2025-10-30

**Soundness:** 3
**Presentation:** 3
**Contribution:** 3
**Rating:** 6
**Confidence:** 3

**Summary:**

PANICL introduces a technically simple but novel inference-time strategy to mitigate over-reliance on single prompts in Visual In-Context Learning.

**Strengths:**

- While smoothing and retrieval are known techniques, PANICL uniquely applies them at the patch/token-level on decoder logits, using score pooling within the VQGAN codebook space.

- PANICL requires no fine-tuning, no retraining, and no model changes, yet shows consistent gains across multiple VICL backbones.

- Broad, well-controlled experiments on several tasks, e.g. segmentation, detection, colorisation, etc.

**Weaknesses:**

- Some components are adapted from broader ML ideas, but their particular application and integration here is new in VICL.

- Lack of theoretical analysis of the method.

- compared to single in-context learning, there is computational overhead from multiple prompt passes.

While the idea is intuitive and shows solid empirical improvements across several VICL backbones and tasks, the methodological contribution is relatively limited, as it largely repurposes known techniques like score averaging and prompt retrieval without substantial technical innovation. The paper is well-executed and practical, but the novelty and depth may not fully meet the bar for a strong ICLR contribution. I find myself between a 4 and a 6, but leaning generous due to the broad applicability and clean results.

**Questions:**

- Why does PANICL in Table 1 show diminishing returns as the number of prompts m increases from 2 to 7? Shouldn’t more prompts consistently improve performance?

---

> ### Author Response · Authors · 2025-11-21
> **Response to Reviewer guEZ (Part 1, Q1 and Q2)**
>
> We sincerely thank the Reviewer ``guEZ`` for the encouraging assessment and constructive comments, which have greatly contributed to strengthening the analysis and methodological depth. Below, we address the questions and comments in detail.
> ***
> >**Q1: Methodological contribution.**\
> *…the methodological contribution is relatively limited, as it largely repurposes known techniques like score averaging and prompt retrieval without substantial technical innovation.*
>
> **A1:** First, thank you for acknowledging our contribution in VICL. We appreciate the concern about methodological contribution and would like to clarify it. PANICL is introduced as a training-free, general framework for VICL models that explicitly formulates and analyzes patch-level, neighbor-aware assignment score smoothing across multiple prompts, which, to the best of our knowledge, has not been explored in prior VICL work.
>
> Our main focus is to mitigate single-prompt reliance, a well-known limitation in VICL, and to understand how multiple prompts can be exploited at the intermediate assignment score level. We respectfully clarify PANICL's methodological contribution as follows:
>
> - Patch-level, neighbor-aware smoothing.
>   Existing multi-example VICL methods (e.g., Large Canvas, Query Voting) either arrange multiple in-context pairs on a single canvas or aggregate at the output pixel level. Ensemble methods such as Feature Ensemble average features at each attention layer, treating all prompts equally. In contrast, PANICL operates directly on intermediate assignment scores at the patch level: for each query patch, it selects $k$-nearest neighbors in the assignment score (or feature) space across prompts and aggregates them with similarity-based weights (JS divergence or $\ell_2$ distance). This patch-level, neighbor-aware assignment score smoothing has not been considered in prior VICL works.
>
> - Prompt pool construction for multi-prompt VICL.
>   While retrieving globally similar in-context pairs for a query has been explored in previous work, constructing an effective prompt pool to integrate multiple prompts under single-prompt reliance remains underexplored. PANICL explicitly builds a prompt pool by combining the query with each retrieved in-context pair as separate visual prompts, and then performs patch-level smoothing over the assignment scores from this pool. This provides an effective mechanism for leveraging multiple prompts.
>
> - A general method transferable across common VICL models.
>   To the best of our knowledge, prior work has not proposed and validated a single assignment score smoothing method that can be instantiated across different types of VICL models. In PANICL, we explicitly design how to adapt the same patch-level, neighbor-aware smoothing to different representations: discrete codebook logits in MAE-VQGAN and LVM, and continuous pixel-space features in SegGPT and Painter. This requires non-trivial methodological decisions beyond reusing the exact same implementation. We then validate this general method on diverse tasks (foreground and multi-object segmentation, detection, colorization, keypoint detection), showing that the same smoothing mechanism consistently extends across VICL models.
>
> In summary, we respectfully argue that systematically formulating patch-level, neighbor-aware assignment score smoothing, constructing an effective prompt pool for multi-prompt VICL, and validating its transferability across diverse VICL models and tasks, constitutes a meaningful and non-trivial methodological contribution.
> ***
> >**Q2: Lack of theoretical analysis.**\
> *Lack of theoretical analysis of the method.*
>
> **A2:** We appreciate the constructive comment regarding theoretical analysis. Our current paper primarily focuses on an empirical investigation of our central assumption: that assignment scores from a single in-context pair are overly specialized, and that smoothing them with scores from neighboring pairs effectively reduces this bias.
>
> Concretely, PANICL can be viewed through a simple and intuitive lens: for each patch, the assignment score (or feature) from a single in-context pair provides a biased estimate of the ideal score aligned with the ground truth. PANICL performs patch-level, neighbor-aware smoothing of these estimates across prompts, which mitigates the bias induced by using a single prompt in the assignment score space.
>
> A fully rigorous theoretical analysis is beyond the scope of this work, and we view it as an exciting direction for future research. In the revised version, we have added a dedicated discussion of this perspective and its implications in Appendix Section F; please kindly refer to it.

---

> ### Author Response · Authors · 2025-11-21
> **Response to Reviewer guEZ (Part 2, Q3 and Q4)**
>
> >**Q3: Computational overhead compared to single in-context learning.**\
> *Compared to single in-context learning, there is computational overhead from multiple prompt passes.*
>
> **A3:** Thank you for the comment on computational cost. We acknowledge that PANICL introduces additional computational overhead because it passes multiple prompts per query and constructs a prompt pool. However, this overhead is moderate and, in practice, marginal compared to the cost of the single-prompt baseline.
>
> In Appendix Section C, we provide a detailed analysis of the computational overhead. Table 18 compares inference time against PLR and Large Canvas, and Table 19 reports the per-prompt resource usage of the Pool. Pool construction adds only 0.025 s per prompt and uses about 3.3 MB of GPU memory and 3.9 MB of cache per prompt. This overhead is negligible relative to the backbone forward pass and can be parallelized across prompts in practice.
> ***
> >**Q4: Diminishing returns when increasing the number of prompts $m$.**\
> *Why does PANICL in Table 1 show diminishing returns as the number of prompts $m$ increases from 2 to 7? Shouldn’t more prompts consistently improve performance?*
>
> **A4:** Thank you for this insightful question. This behavior occurs because, as $m$ increases, the off-the-shelf in-context pair retriever tends to incorporate more irrelevant examples into the prompt pool, which dilutes the benefits of smoothing. This observation is supported by Table 8, where applying smoothing over larger windows or all patches results in performance degradation. Moreover, as detailed in our failure case analysis in Appendix Section E, when the in-context pair retriever introduces low-quality or misleading features, the additional prompts can confuse the VICL model, negatively impacting performance.
>
> We have included a detailed analysis explaining this degradation trend in Section 4.2 of the revised paper.
>
> Once again, we thank Reviewer ``guEZ`` for the constructive feedbacks provided during this review.

---

### Official Review · Reviewer_Ujk5 · 2025-11-04

**Soundness:** 3
**Presentation:** 3
**Contribution:** 2
**Rating:** 4
**Confidence:** 3

**Summary:**

This paper introduces PANICL, a method to fix a key problem in Visual In-Context Learning, or VICL, where models over-rely on a single example prompt, leading to biased and unstable predictions. PANICL is a training-free framework that mitigates this issue by using multiple in-context pairs. For a given query image, it first retrieves several similar example pairs from a support set. It then processes each of these pairs through the VICL model to create a prompt pool of their patch-level assignment scores, which are the model's internal predictions. The query image is also processed to obtain its own initial scores. The core idea is assignment score smoothing, where PANICL finds the k nearest neighbors for the query's scores from the prompt pool using JS divergence and then calculates a weighted average. This smoothing step debiases the query's prediction by incorporating information from multiple examples. The authors show this method consistently improves performance on various tasks like segmentation, detection, and colorization, which is robust to domain shifts and can be generalized to other VICL models such as SegGPT, Painter, and LVM.

**Strengths:**

1. The paper directly addresses a well-known and critical weakness in VICL. The over-reliance on a single prompt is a key bottleneck, and PANICL's approach of using multiple examples is an intuitive and logical solution.
2. Instead of ensembling intermediate features or ensembling final predictions, PANICL operates at the assignment score level. This allows it to correct biases at a patch-by-patch level before the final prediction is rendered, offering a more granular way to debias the model.
3. A major strength is that PANICL is a "training-free" framework. It does not require any model fine-tuning, making it an efficient, plug-and-play improvement.
4. The method shows consistent and significant performance gains over strong baselines, including both single-example methods and other multi-example methods. The gains are not just in accuracy but also in robustness to domain shifts, a critical factor for real-world deployment.

**Weaknesses:**

1. The method's effectiveness is highly dependent on the quality of the retrieved in-context pairs. The authors admit this in their failure case analysis (Section E, Figure 12), showing that when the retriever provides examples that are globally similar but semantically misaligned (e.g., different object scale or position), PANICL's performance can degrade significantly, sometimes even below the single-prompt baseline.
2. The method appears to be heuristic and parameter sensitive, which may require careful tuning for new tasks or datasets.
3. While it improves SegGPT, the gain on foreground segmentation is minor. Similarly, for keypoint detection with Painter, the gain is minimal.
4. Although the method is intriguing, its novelty is limited. I feel this paper is leaning toward a strong engineering method instead of proposing a new framework.

**Questions:**

1. Your method requires m additional forward passes to build the prompt pool. Have you explored cheaper alternatives, such as using a static pool of generic assignment scores rather than a dynamic pool retrieved for every query?
2. You show that patch-level k-NN (Table 8, "PANICL") is better than retrieving neighbors from all patches (Table 8, "All-patch"). This observation is interesting, does forcing the model to find neighbors only in the same spatial location (e.g., top-left) prevent it from getting confused by patches from other parts of the image?
3. You use JS Divergence to compare assignment score distributions. This works for token-based models like MAE-VQGAN and LVM. But for pixel-space models like SegGPT and Painter, you switch to l2 distance on intermediate features. Does this mean your core "assignment score smoothing" idea doesn't apply to these models, and you are instead performing a type of "feature ensemble," similar to the baseline?

I am willing to reconsider my evaluation.

---

> ### Author Response · Authors · 2025-11-21
> **Response to Reviewer Ujk5 (Part 1, Q1 and Q2)**
>
> We thank Reviewer ``Ujk5`` for the constructive comments and valuable suggestions, which have greatly helped us improve the clarity and depth of this work. Below we provide detailed responses to the questions and comments.
> ***
> >**Q1: Cost-efficient alternatives (static prompt pool)**\
> *...Have you explored cheaper alternatives, such as using a static pool of generic assignment scores rather than a dynamic pool retrieved for every query?*
>
> **A1:** This is a very insightful and important question. As suggested, we explored whether a static prompt pool can serve as a cheaper alternative to our per-query dynamic prompt pool.
>
> Concretely, we construct an offline, static, patch-level prompt pool using all samples in the support set $\mathcal{S}$. Since the query is unavailable during such offline construction, each in-context example serves as its own anchor (i.e., we use $\mathcal{P}_l^{\text{self}}$ instead of $\mathcal{P}_l^{\text{q}}$), which yields the best performance in Table 7. The number of neighbors is set to $k = 4$. The results are:
> | Method        | Fold-0 | Fold-1 | Fold-2 | Fold-3 | Mean  |
> |---------------|:------:|:------:|:------:|:------:|:-----:|
> | Static Pool   | 35.71  | 39.05  | 36.62  | 33.42  | 36.20 |
> | **Dynamic Pool** | **38.18** | **40.63** | **37.82** | **35.02** | **37.91** |
>
> The dynamic prompt pool clearly outperforms the static prompt pool. We attribute this to the fact that the static pool cannot leverage query-anchored combinations: it is built without seeing the query, and thus lacks the query-specific alignment that PANICL exploits. Nevertheless, the static pool still benefits from multi-example smoothing and outperforms the single-prompt PLR baseline (even without an in-context retriever), due to our proposed assignment score smoothing.
>
> In the revised paper, we have added this comparison in Section 4.7 (Table 9). Please kindly refer to it.
> ***
> >**Q2: Effect of spatial locality in patch-level $k$-NN retrieval**\
> *...This observation is interesting, does forcing the model to find neighbors only in the same spatial location (e.g., top-left) prevent it from getting confused by patches from other parts of the image?*
>
> **A2:** Thank you for this insightful question about the role of spatial locality in our patch-level $k$-NN retrieval. Table 8 shows that spatially aligned, per-patch retrieval (our PANICL setting) is more effective than the all-patch setting. To further investigate this observation, we conducted additional experiments varying the spatial retrieval window size.
>
> Specifically, for each query patch, we define a local spatial window (e.g., $2{\times}2$, $3{\times}3$, or $5{\times}5$) on the $7{\times}7$ patch grid as the search region for neighbor retrieval. The maximum window size is the entire $7{\times}7$ grid (the All-patch setting), and the minimum window size is $1{\times}1$ (our PANICL setting, i.e., per-patch retrieval). When a window would extend beyond the image boundary, we clamp its top-left corner to the last valid position so that the entire window remains inside the grid.
>
> We compare different window sizes as follows:
> | Window size | Fold-0 | Fold-1 | Fold-2 | Fold-3 | Mean  |
> |----------------------|:------:|:------:|:------:|:------:|:-----:|
> | 2 × 2                | 37.34  | 39.98  | 37.37  | **35.02** | 37.40 |
> | 3 × 3                | 37.01  | 39.76  | 37.08  | 34.70  | 37.14 |
> | 5 × 5                | 36.73  | 39.66  | 36.95  | 34.56  | 36.98 |
> | All-patch (7 × 7)    | 36.72  | 39.64  | 36.97  | 34.59  | 36.98 |
> | **PANICL (1 × 1)**   | **38.18** | **40.63** | **37.82** | **35.02** | **37.91** |
>
> PANICL ($1{\times}1$ window) achieves the highest mean mIoU and consistently outperforms larger-window settings across all folds. Moreover, performance monotonically degrades as the window size increases, i.e., as more spatially distant patches are allowed as neighbors.
>
> These results indicate that strict spatial locality is critical: forcing neighbors to lie at the same spatial location (or in a very small neighborhood) reduces cross-region confusion and yields better smoothing. In other words, the gains of PANICL over the all-patch setting are not only due to assignment score smoothing itself, but also to the structured way in which we select neighbors in the same spatial region, which is also a key aspect of our contribution.
>
> We have added the additional results and analysis in Section 4.7 (Table 8). Please kindly refer to it.

---

> ### Author Response · Authors · 2025-11-21
> **Response to Reviewer Ujk5 (Part 2, Q3 and Q4)**
>
> >**Q3: Applicability of assignment score smoothing vs. feature ensemble on pixel-space models.**\
> *You use JS Divergence to compare assignment score distributions… But for pixel-space models like SegGPT and Painter, you switch to l2 distance on intermediate features. Does this mean your core "assignment score smoothing" idea doesn't apply to these models, and you are instead performing a type of "feature ensemble," similar to the baseline?*
>
> **A3:** Thank you for this important question. Our original motivation was that token-based VICL models (e.g., MAE-VQGAN, LVM) naturally provide discrete assignment score distributions in the codebook space, for which JS divergence is a principled comparison metric. For pixel-space models such as SegGPT and Painter, we do not have the same discrete assignment scores, so we initially used $\ell_2$ distance on normalized intermediate features as a proxy similarity measure.
>
> Following your suggestion, we explicitly tested whether the core idea of assignment score smoothing still applies to SegGPT by comparing different similarity measures (PLR + FE vs. $\ell_2$ distance vs. JS divergence). On the foreground segmentation task, we obtained:
>
> | Method | Fold-0 | Fold-1 | Fold-2 | Fold-3 | Mean  |
> |----------------------|:------:|:------:|:------:|:------:|:-----:|
> | PLR + FE             | 69.44  | 77.26  | 79.06  | 77.77  | 75.88 |
> | PANICL ($\ell_2$)       | 69.68  | 77.50  | 79.29  | 78.05  | 76.13 |
> | PANICL (JS)          | 69.76  | 77.44  | 79.27  | 77.95  | 76.11 |
>
> Both similarity measures ($\ell_2$ and JS) consistently improve over PLR + FE, and the differences between $\ell_2$ and JS is marginal for each fold. This suggests that our claim on assignment score smoothing is still beneficial for pixel-space VICL models, even though the underlying representation is not a discrete codebook.
>
> In practice, for pixel-space VICL models, we first normalize the intermediate features so that each patch can be interpreted as a distribution over latent dimensions; JS divergence then measures the discrepancy between these normalized responses analogously to token-based assignment scores. Because both JS and $\ell_2$ are applied to the same normalized patch representations, they induce very similar nearest-neighbor rankings, which explains why their performance is nearly identical. These observations further support our claim that assignment score smoothing is applicable to pixel-space VICL models.
>
> The difference between FE and PANICL is as follows: FE conducts unconditional averaging across multiple examples, whereas PANICL performs neighbor-aware smoothing at the patch level.
>
> In the revision, we have added the JS divergence results on SegGPT in Appendix Section A (Table 17). Please kindly refer to that table.
> ***
> >**Q4: Dependence on retrieval quality and semantic alignment of in-context pairs.**\
> *The method's effectiveness is highly dependent on the quality of the retrieved in-context pairs…*
>
> **A4:** We appreciate this insightful comment. We agree that the retriever plays an important role in VICL performance in general, and PANICL is not completely immune to poor retrieval, especially when examples are globally similar but semantically misaligned.
>
> To stress-test this aspect, we conducted an experiment in Appendix Section A (Table 14), where we intentionally forced the in-context retriever to return examples whose class labels differ from those of the query in each fold. In other words, the retrieved examples are globally similar but semantically misaligned by construction. We compare PANICL against PLR (the single-prompt baseline) and report mIoU on the FgSeg. task. Even under this severe setting, PANICL consistently outperforms PLR across all folds.
>
> These results show that while retrieval quality certainly affects the absolute performance, PANICL still provides a robust improvement over single-prompt VICL in challenging scenarios.

---

> ### Author Response · Authors · 2025-11-21
> **Response to Reviewer Ujk5 (Part 3, Q5 and Q6)**
>
> >**Q5: Heuristic and parameter-sensitive, requiring careful tuning.**\
> *The method appears to be heuristic and parameter sensitive, which may require careful tuning for new tasks or datasets.*
>
> **A5:** We thank the reviewer for pointing this out. PANICL does introduce a few hyper-parameters (e.g., the number of prompts $m$, the number of neighbors $k$, the temperature $\tau$, and the balance factor $\alpha$). However, in practice we find that PANICL is reasonably robust and does not require careful per-task tuning to provide gains.
>
> For example, with MAE-VQGAN, we use the same default settings across multiple tasks and datasets:
> - For fine-grained tasks (e.g., segmentation, and colorization), we use smaller $m$ and $k$ (e.g., 3 or 4) and default $\tau$ and $\alpha$ (e.g., $\tau{=}1.0$, $\alpha{=}1.0$).
> - For coarse-grained tasks (e.g., object detection), we use slightly larger $m$ and $k$ (e.g., 5 or 6) and a smaller $\alpha$ (e.g., 0.7 or 0.8) with the same default $\tau$ (e.g., 1.0).
>
> Importantly, we reuse these hyper-parameter settings on new datasets (e.g., COCO) and on novel out-of-domain classes (e.g., FSS-1000) without additional tuning, and still observe consistent improvements over other baselines. This suggests that PANICL is not overly sensitive to hyper-parameters and that simple heuristic defaults already work well in practice.
>
> To address this insightful point, we have added a short "practical guidance" paragraph in Appendix Section A to summarize these recommended default settings and to further clarify that extensive hyper-parameter tuning is not required. Please kindly refer to it.
> ***
> >**Q6: Limited improvements on SegGPT and Painter.**\
> *While it improves SegGPT, the gain on foreground segmentation is minor. Similarly, for keypoint detection with Painter, the gain is minimal.*
>
> **A6:** We thank the reviewer for raising this question. We acknowledge that on foreground segmentation the performance gain is less pronounced in quantitative measurements (Table 5) compared to qualitative observations (Figure 3). Our main focus, however, is on mitigating single-prompt reliance in VICL models via a training-free strategy. From this perspective, we believe the results on SegGPT and Painter are still meaningful for two reasons:
> - SegGPT is already a highly competitive VICL backbone. Even small numerical improvements on such a strong baseline, obtained without any additional training or architectural changes, indicate that PANICL can reliably enhance existing models rather than relying on weaker starting points.
> - For keypoint detection with Painter, our PANICL results are on par with the learnable prompting strategy reported in the Painter paper, while remaining completely training-free. This suggests that assignment score smoothing can effectively bridge the gap to learned prompt optimization.

---

> ### Author Response · Authors · 2025-11-21
> **Response to Reviewer Ujk5 (Part 4, Q7)**
>
> >**Q7: Limited novelty; primarily an engineering contribution.**\
> *Although the method is intriguing, its novelty is limited. I feel this paper is leaning toward a strong engineering method instead of proposing a new framework.*
>
> **A7:** We appreciate the reviewer's candid assessment regarding novelty. PANICL is proposed as a training-free, general framework for VICL that explicitly proposes and analyzes patch-level, neighbor-aware assignment score smoothing across multiple prompts, which, to the best of our knowledge, has not been explored in prior VICL work.
>
> Our research mainly focuses on mitigating single-prompt reliance in VICL and exploring how to effectively exploit multiple prompts through assignment score smoothing. PANICL introduces a patch-level, neighbor-aware assignment score smoothing mechanism across multiple prompts in the VICL setting, and we demonstrate its effectiveness and broad applicability across multiple VICL models and downstream tasks. Below, we clarify how PANICL differs from existing multi-example and ensemble methods:
>
> - Multi-example VICL methods (Large Canvas, Query Voting).\
> Large Canvas arranges multiple examples on a single canvas and feeds this combined canvas into the VICL model in a single forward pass, sacrificing image details due to resizing. It does not perform any smoothing at the intermediate assignment score level. In contrast, PANICL keeps each example on its own canvas, runs the VICL model on each example separately, and then smooths the intermediate assignment scores across prompts from a patch-level, neighbor-aware perspective. This design avoids pixel-level compromises caused by arranging multiple examples on a single canvas and allows the inputs to be processed at their original resolution.\
> Query Voting ensembles predictions at the output pixel level, using thresholds on pixel values to decide foreground or background. This makes it difficult to extend beyond foreground segmentation. PANICL instead operates on intermediate assignment scores, rather than on output pixels, which allows it to be applied to a wider range of tasks beyond foreground segmentation, such as colorization, multi-object segmentation, and keypoint detection.
>
> - Ensemble methods (Feature Ensemble).\
> Feature Ensemble simply averages features from multiple examples at each attention layer, treating all prompts equally, and is often tailored to specific architectures. PANICL, by contrast, performs neighbor-aware, weighted smoothing at the patch level: it selects $k$-nearest neighbors in the intermediate assignment-score (or feature) space and aggregates them with similarity-based weights (via JS divergence or $\ell_2$ distance), introducing a patch-level, neighbor-aware assignment score smoothing mechanism that is not present in prior VICL ensemble methods.
>
> In summary, while PANICL is implementation-friendly, we respectfully argue that its novelty lies in targeting the patch-level assignment scores for smoothing. This approach effectively addresses the limitations of prior strategies that operate primarily at the input or output levels. We therefore consider the systematic formulation and validation of this neighbor-aware smoothing mechanism to be a meaningful and non-trivial contribution beyond a minor engineering tweak.
>
> In the revised paper, we have further highlighted the novelty of PANICL and clarified its differences from previous approaches in Sections 1 and 2; please kindly refer to these sections.
>
> Last but not least, we sincerely thank Reviewer ``Ujk5`` again for the valuable comments provided during this review.

---

> ### Comment · Reviewer_Ujk5 · 2025-11-28
>
> Thanks for the detailed rebuttal. However, I still find that the novelty of your work is incremental, so I will maintain my score.

---

> ### Author Response · Authors · 2025-12-04
> **Further Clarification on Novelty and Technical Contribution**
>
> We thank the Reviewer ``Ujk5`` for the follow-up. We understand the concern that the method might appear incremental due to its simplicity. However, we respectfully argue that **simplicity in execution should not be conflated with incrementalism in insight.**
>
> We would like to emphasize that our contribution is **exploratory and foundational** rather than just an engineering tweak. To the best of our knowledge, this work is the **first** to:
> - **Identify the bias in a novel space:** Explicitly demonstrate that single-example bias appears as deviations in the assignment score space.
> - **Propose a general mechanism:** Formulate patch-level, neighbor-aware smoothing as the general solution to this specific bias.
> - **Dynamic construction:** Introduce a **dynamic prompt pool** strategy that actively selects high-quality query-anchor combinations, distinct from static or random baselines.
>
> Different from prior methods, our novelty lies in identifying that the **assignment score space** is the general and effective level for aggregation across diverse VICL models and tasks.
>
> We believe that identifying the correct representation space and proposing a general, training-free framework to solve a critical bottleneck constitutes a meaningful methodological step forward for the VICL community.

---

### Author Response · Authors · 2025-11-21
**General Response**

We sincerely thank our reviewers for their time and effort in providing thorough and constructive feedback on our paper.
***
We are encouraged to see that our reviewers recognize this work as a **novel, effective, and broadly applicable inference-time strategy** for mitigating single-prompt reliance in Visual In-Context Learning (VICL):
- Reviewers ``Ujk5``, ``guEZ``, and ``rqcE`` acknowledge the empirical strength of our approach, noting that it achieves *"consistent and significant performance gains,"* *"consistent gains across multiple VICL backbones"* and is *"quite effective."*
- Reviewers ``Ujk5``, ``guEZ``, and ``rqcE`` commend our method's *"robustness to domain shifts,"* ability to generalize to *"multiple VICL backbones,"* and the *"clean results"* on several benchmarks.
- Reviewers ``Ujk5`` and ``guEZ`` highlight the novelty of our specific design, stating that PANICL is *"new in VICL,"* *"uniquely applies"* smoothing at the *"patch/token-level"* and offers a *"more granular way to debias the model"* compared to prior methods.
- Reviewers ``guEZ`` and ``rqcE`` appreciate the clarity of our work, describing the paper as *"well-executed,"* *"practical,"* and *"fairly well written."*
- Reviewers ``Ujk5`` and ``rqcE`` think our method addresses a *"well-known and critical weakness in VICL,"* and an *"important problem"* with an *"intuitive, logical, and interesting"* idea.
- Reviewers ``Ujk5`` and ``guEZ`` highlight a *"major strength"* in PANICL's *"training-free"* nature, noting that it requires *"no fine-tuning"* or *"model changes,"* making it an efficient, *"plug-and-play improvement."*
***
**We highlight our contributions as follows:**

To the best of our knowledge, PANICL is the **first** framework to explicitly identify that single-prompt bias manifests systematically in the **assignment score space** and to propose a general solution. Specifically:
- We introduce a **dynamic prompt pool** that combines the query with retrieved examples to construct high-quality neighbors.
- We operate explicitly in the **assignment score space,** identifying it as the effective aggregation level for diverse VICL models.
- We perform **patch-level, neighbor-aware smoothing** via similarity-based weights (JS divergence or $\ell_2$ distance).

As a training-free framework, PANICL demonstrates consistent improvements across diverse VICL models (e.g., MAE-VQGAN, SegGPT, Painter, LVM) and downstream tasks without fine-tuning, retraining, or model changes.
***
As suggested by the reviewers, we revised our manuscript (changes highlighted in orange) as follows:

***Additional Experiments and Analyses:***
- Added a comparison with a static prompt pool in **Section 4.7 (Table 9)** to demonstrate the necessity of our dynamic retrieval strategy, as suggested by Reviewer ``Ujk5``.
- Added an ablation study on spatial retrieval window sizes in **Section 4.7 (Table 8)** to validate the effectiveness of per-patch retrieval, as suggested by Reviewer ``Ujk5``.
- Added JS divergence results on SegGPT in **Appendix A (Table 17)** to confirm the applicability of our assignment score smoothing mechanism on pixel-space models, as suggested by Reviewer ``Ujk5``.
- Added detailed results of per-class retrieval quality vs. performance in **Appendix A (Tables 15 & 16),** addressing concerns from Reviewer ``rqcE``.
- Added an analysis explaining the performance saturation/degradation with increased prompt numbers ($m$) in **Section 4.2,** as suggested by Reviewers ``guEZ`` and ``rqcE``.

***Textual Revisions and Clarifications:***
- Expanded **Sections 1 and 2** to explicitly clarify the differences between PANICL and prior methods (e.g., Large Canvas, Query Voting, and Feature Ensemble), addressing the novelty concerns from Reviewers ``Ujk5`` and ``rqcE``.
- Added a discussion on the robust improvements on SegGPT for the FgSeg. task in **Section 4.5,** in response to comments from Reviewers ``Ujk5`` and ``rqcE``.
- Added a discussion on the theoretical perspective of PANICL in **Appendix F,** as suggested by Reviewer ``guEZ``.
- Added a *"Practical Guidance"* paragraph in **Appendix A** to summarize recommended default settings, as suggested by Reviewer ``Ujk5``.
- Revised the claim from *"model-agnostic"* to *"general"* in Section 1, as suggested by Reviewer ``rqcE``.
- Corrected the boldface entries in Table 1, as pointed out by Reviewer ``rqcE``.

---

### Author Response · Authors · 2025-12-04
**Summary Comment for the New Area Chairs**

We sincerely thank the Area Chairs for handling our paper. We take this opportunity to briefly summarize our work, the main reviewer comments, and our responses.
***
## Paper Summary
This paper addresses a critical weakness in Visual In-Context Learning (VICL): the strong over-reliance on a single in-context example, which often leads to biased and unstable predictions. We propose PANICL, a training-free and general framework to mitigate this bias. Instead of operating at the input or output level, PANICL builds a dynamic prompt pool and performs patch-level, neighbor-aware smoothing explicitly in the assignment score space.

PANICL is evaluated on four VICL backbones (MAE-VQGAN, SegGPT, Painter, LVM) and five downstream tasks, yielding consistent improvements over strong single-prompt and multi-prompt baselines.
***
## Reviews and Responses
Overall, reviewers acknowledged the importance of the problem and the conceptual merit of PANICL. They appreciated that it directly targets single-prompt over-reliance, introduces patch-level smoothing in the assignment score space, and remains training-free while showing consistent gains.

In summary, the main reviewer comments and our responses are as follows (all changes are reflected in the revised manuscript, highlighted in orange):
### (a) Novelty and technical contribution
Our work is exploratory in nature. To the best of our knowledge, it is the **first** to (i) explicitly demonstrate that single-prompt VICL exhibits systematic bias **in the assignment score space**, and (ii) show that smoothing assignment scores across multiple prompts can effectively mitigate this bias. The core methodological contribution is **patch-level, neighbor-aware assignment score smoothing,** formulated as a **general** mechanism for VICL models and tasks. We also provide detailed analyses on **why** and **how** this idea works:
- **Why:** Biased predictions from different single-prompt examples appear as deviations in the assignment score space. Patch-level smoothing enforces spatial locality across prompts, while similarity-based weighting naturally allows compatible neighbors to contribute more, leading to reduced bias and stable predictions.
- **How:** We are the first to
  1. Propose a **dynamic prompt pool** for constructing high-quality neighbors, exploring how different query–anchor combinations affect the pool;
  2. Operate explicitly in **assignment score space** as a general aggregation level for VICL (avoiding the resizing issues of input-level methods and the task limitations of output-level methods);
  3. Perform **patch-level, neighbor-aware smoothing across prompts** via $k$-nearest neighbors with similarity-based weights (JS divergence or $\ell_2$ distance).

Extensive experiments on four VICL backbones and five downstream tasks empirically support these claims. We have further highlighted these distinctions in **Sections 1 and 2.**
### (b) Dependence and sensitivity to retrieval quality
We added detailed results of per-class retrieval quality vs. performance in **Appendix A (Tables 15 & 16),** confirming that while retrieval quality affects absolute performance generally, PANICL consistently outperforms baselines.
### (c) Spatial locality vs. all-patch retrieval
We have added an ablation study on spatial retrieval window sizes in **Section 4.7 (Table 8),** validating the effectiveness of our strict per-patch retrieval.
### (d) Static vs. dynamic prompt pool
In **Section 4.7 (Table 9),** we added a static prompt pool baseline, demonstrating the necessity of our query-anchored dynamic retrieval strategy for prompt pool construction.
### (e) Applicability to pixel-space VICL
We added experiments in **Appendix A (Table 17)** comparing $\ell_2$ and JS divergence on SegGPT, confirming that our assignment score smoothing mechanism generalizes effectively to pixel-space models.
### (f) Hyper-parameter sensitivity
In **Appendix A,** we added a **Practical Guidance** paragraph summarizing recommended default settings and demonstrating their robustness across datasets.
### (g) Lack of theoretical analysis
In **Appendix F**, we added a conceptual discussion interpreting PANICL as patch-level smoothing of biased estimators.
### (h) Performance saturation with increasing prompt numbers ($m$)
In **Section 4.2,** we added an analysis explaining that performance saturation with larger prompt numbers ($m$) stems from noise introduced by the in-context pair retriever.
### (i) Computational overhead
In **Appendix C,** we provided a detailed breakdown showing that prompt pool construction overhead is negligible ($\sim$0.025s per prompt).
### (j) Additional clarification
- Discussion on the robust improvements on SegGPT for the FgSeg. in **Section 4.5.**
- Revised the claim from "model-agnostic" to "general" in **Section 1.**
- Corrected the boldface entries in **Table 1.**
***
We believe the paper has been substantially strengthened. Thank you again for your time and consideration.

---

### Meta-Review · Area_Chair_wov6 · 2025-12-19

**Summary:**

The paper received three expert reviews. Overall the reviews are detailed and informative.

Reviewer rqcE gave the rating of 4 (marginally below acceptance). The reviewer liked the simplicity of the idea and appreciated the results. The main concern from this reviewer is the novelty, which as described, is "minimal". The reason is that the smoothing design is a minor change compared to previous work, and the improvement is too small to justify the contribution. The reviewer questioned why the performance declines with increased m (no. of in-context pairs) and requested elaboration on how the quality of retrieval affects the proposed approach.

Reviewer guEZ gave the rating of 6 (marginally above acceptance). The reviewer recognized the training-free design of the approach but raised a similar concern as the first reviewer, i.e., "methodological contribution is relatively limited" because the method just "repurposes known techniques like score averaging and prompt retrieval". The reviewer was also concerned about lack of theoretical analysis and the computational overhead caused by the use of multiple in-context pairs.

Reviewer Ujk5 gave the rating of 4 (marginally below acceptance). This reviewer also recognized the training-free design but pointed out that the novelty is limited. The reviewer also expressed concerns that the approach has heavy dependence on the retrieval system and may be sensitive to the selection of parameters, which require further tuning for new tasks. The reviewer also mentioned that the gains are minor, especially on foreground segmentation and keypoint detection.

Most concerns seem to be addressed in the rebuttal, except the novelty issue.

**Reviewer Concerns:**

Most reviewers questioned why the performance declines with increased number of in-context pairs. The rebuttal explained that increasing in-context pairs "tends to include more irrelevant examples" and emphasized the importance of the retrieval system. This answer makes sense. Other answers to questions like heavy dependence on retrieval and sensitivity to parameters are also reasonable. However, the concern about lack of novelty remains unaddressed.

**Reviewer Scores:**

By carefully examining the paper, reviews, and rebuttal, the AC has the same view with the reviewers that the novelty of this work is limited. The smoothing design does not fundamentally change the framework and the multi-prompt idea has already been investigated in the literature. The insights offered by this work are not significant enough. Reviewer Ujk5 mentioned after rebuttal that he/she maintained the rating of 4 due to the concern on novelty. Reviewer rqcE is also unlikely to change the rating. Reviewer guEZ's view is more negative than positive. In summary, all reviewers shared the concern on novelty and therefore the scores are negative. The AC recommends rejection.

---

### Decision · Program_Chairs · 2026-01-26

Reject